# Mammalian cell growth dynamics in mitosis

**Teemu P Miettinen[1,2]†\*, Joon Ho Kang[2,3]†, Lucy F Yang[2], Scott R Manalis[2,4,5]\***

[1]MRC Laboratory for Molecular Cell Biology, University College London, London, United Kingdom; [2]Koch Institute for Integrative Cancer Research, Massachusetts Institute of Technology, Cambridge, United States; [3]Department of Physics, Massachusetts Institute of Technology, Cambridge, United States; [4]Department of Biological Engineering, Massachusetts Institute of Technology, Cambridge, United States; [5]Department of Mechanical Engineering, Massachusetts Institute of Technology, Cambridge, United States

**Abstract** The extent and dynamics of animal cell biomass accumulation during mitosis are unknown, primarily because growth has not been quantified with sufficient precision and temporal resolution. Using the suspended microchannel resonator and protein synthesis assays, we quantify mass accumulation and translation rates between mitotic stages on a single-cell level. For various animal cell types, growth rates in prophase are commensurate with or higher than interphase growth rates. Growth is only stopped as cells approach metaphase-to-anaphase transition and growth resumes in late cytokinesis. Mitotic arrests stop growth independently of arresting mechanism. For mouse lymphoblast cells, growth in prophase is promoted by CDK1 through increased phosphorylation of 4E-BP1 and cap-dependent protein synthesis. Inhibition of CDK1-driven mitotic translation reduces daughter cell growth. Overall, our measurements counter the traditional dogma that growth during mitosis is negligible and provide insight into antimitotic cancer chemotherapies.
DOI: https://doi.org/10.7554/eLife.44700.001

**\*For correspondence:**
teemu@mit.edu (TPM);
srm@mit.edu (SRM)

†These authors contributed equally to this work

## Introduction

Animal cell growth, that is biomass accumulation (*Lloyd, 2013*), is classically viewed to take place during interphase. During mitosis and cytokinesis, when cells are assumed to prioritize their energy usage for executing cell division, growth is presumed to be minimal (reviewed in *Kronja and Orr-Weaver, 2011*; *Pyronnet and Sonenberg, 2001*; *Salazar-Roa and Malumbres, 2017*; *Sivan and Elroy-Stein, 2008*; *White-Gilbertson et al., 2009*). More specifically, mRNA synthesis is inhibited due to chromatin condensation and dissociation of transcription factors (*Liang et al., 2015*; *Novais-Cruz et al., 2018*; *Parsons and Spencer, 1997*; *Prescott and Bender, 1962*), and ribosomal RNA synthesis is blocked as the nucleolus disappears in prometaphase (*Hernandez-Verdun, 2011*). Protein synthesis has also been reported to be suppressed in cell populations enriched for mitosis (*Bonneau and Sonenberg, 1987*; *Celis et al., 1990*; *Fan and Penman, 1970*; *Prescott and Bender, 1962*; *Pyronnet et al., 2001*; *Sivan et al., 2011*; *Sivan et al., 2007*). Consistently, polysome and ribosome profiling studies have suggested that the translational efficiency of most mRNAs is reduced in mitosis (*Park et al., 2016*; *Qin and Sarnow, 2004*; *Stumpf et al., 2013*; *Tanenbaum et al., 2015*). Studies on individual components of the translational machinery, such as eukaryotic Translation Initiation Factor 4E (eIF4E), eIF4E Binding Protein 1 (4E-BP1), eukaryotic Translation Elongation Factor 2 (eEF2) and S6 Ribosomal Protein (S6RP) have also suggested reduced protein synthesis, especially cap-dependent translation initiation, in mitosis (*Celis et al., 1990*; *Dobrikov et al., 2014*; *Pyronnet et al., 2001*; *Shah et al., 2003*; *Wilker et al., 2007*).

Furthermore, ribosomes disassociate from endoplasmic reticulum around metaphase, suggesting that translation may be limited in the middle of mitosis (*Puhka et al., 2007*).

However, this classical view that growth is inhibited during mitosis has recently been challenged. Parts of DNA remain accessible for transcription machinery and de novo transcription of genes involved in cell growth persists in mitosis (*Chan et al., 2012*; *Chen et al., 2005*; *Liu et al., 2017*; *Palozola et al., 2017*). Recent reports also suggest that protein synthesis may persist during mitosis (*Coldwell et al., 2013*; *Shuda et al., 2015*; *Stonyte et al., 2018*). Importantly, cyclin-dependent kinase 1 (CDK1), the key regulator of mitotic entry and progression (*Diril et al., 2012*; *Gavet and Pines, 2010*), phosphorylates and activates components of the protein synthesis machinery, including 4E-BP1 (*Heesom et al., 2001*; *Jansova et al., 2017*; *Shuda et al., 2015*), eEF2 kinase (*Smith and Proud, 2008*) and p70 S6 kinase (*Papst et al., 1998*), suggesting an activation of cap-dependent translation. In addition, cap-independent translation of many mRNAs remains active in mitosis (*Cornelis et al., 2000*; *Marash et al., 2008*; *Pyronnet et al., 2000*; *Qin and Sarnow, 2004*). It is therefore becoming evident that particular proteins, especially those required for completion of cell division and those critical for cell growth, are synthesized during mitosis (*Aviner et al., 2013*; *Aviner et al., 2017*; *Cornelis et al., 2000*; *Marash et al., 2008*; *Park et al., 2016*; *Pyronnet et al., 2000*; *Stumpf et al., 2013*; *Tanenbaum et al., 2015*). However, the extent and dynamics of protein synthesis in mitosis remain unclear.

Importantly, protein and RNA synthesis rates are only proxies of overall growth (biomass increase), which is determined by the balance between synthesis (anabolic) and degradation (catabolic) rates (*Lloyd, 2013*; *Miettinen and Björklund, 2015*; *Miettinen et al., 2017*). Overall growth behavior during mitosis has not been studied, primarily due to the lack of precise cell size measurement methods that are sensitive enough to quantify growth during the short mitotic stages. Here, we utilize suspended microchannel resonator (SMR), a high-precision microfluidic mass sensor, and protein synthesis assays in conjunction with cell cycle measurements to study the extent, dynamics, mechanisms and consequences of mitotic growth on a single-cell level.

## Results

### Animal cells grow during mitosis and cytokinesis

SMR is a microfluidic cantilever that is capable of measuring buoyant mass (a proxy of dry mass, referred to as mass from here on) of single cells with a precision of <0.1 pg (*Figure 1a*; *Figure 1— figure supplement 1a–d*) (*Burg et al., 2007*; *Son et al., 2015a*; *Son et al., 2012*). This resolution corresponds to <8 nm (<0.07%) change in a spherical lymphocyte cell diameter. We repeatedly measured mass of the same cell every ~1 min, resulting in a temporal resolution of approximately 2 min according to the Nyquist rate. We quantified growth, more specifically mass accumulation, throughout multiple cell cycles in L1210 mouse lymphocytes without perturbing normal growth rates (*Figure 1b*) (*Son et al., 2015a*; *Son et al., 2012*). We assigned approximate mitotic entry (i.e. G2/M transition), metaphase-to-anaphase transition (i.e. M/A transition) and cytokinetic abscission of the daughter cells for each cell using biophysical properties and FUCCI cell cycle reporter (*Figure 1— figure supplement 2*; Materials and methods) (*Kang et al., 2019*). This allowed quantification of mass accumulation during early mitosis (between G2/M transition and M/A transition) and cytokinesis (between M/A transition and daughter cell abscission) on a single-cell level (*Figure 1c*). In cytokinesis the elongated cells register smaller than round cells in our mass measurements, because of a change in mass distribution (*Kang et al., 2019*). Correcting for this cell elongation induced bias (correction is applied to all data shown, unless otherwise stated) (see Materials and methods), had little influence of the mass measurements during cytokinesis (*Figure 1—figure supplement 1e*).

In total, we analyzed 180 individual L1210 cells undergoing mitosis and observed that on average 12% of the total mass accumulated during the whole cell cycle was acquired during M-phase (i.e. during mitosis and cytokinesis) (*Figure 1d*; *Figure 1—source data 1*). 7% of total cell growth took place during early mitosis, while 5% took place during cytokinesis. During anaphase, duration of which was estimated based on cell elongation (Materials and methods), mass accumulation was negligible. Considering that in most cell lines M-phase lasts approximately 10% of the whole cell cycle, the 12% mass accumulation observed during M-phase makes M-phase growth comparable to interphase growth.

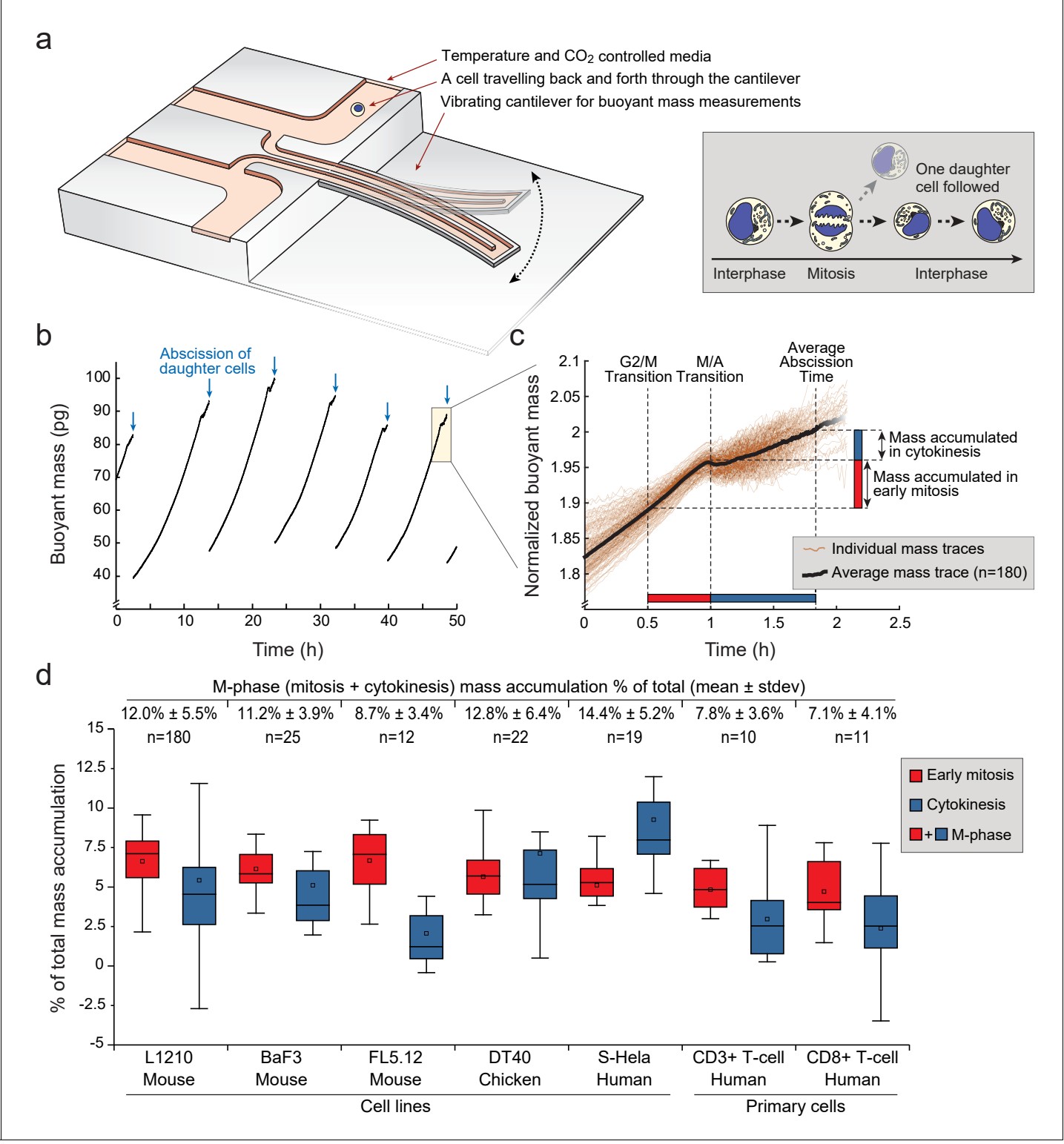

**Figure 1.** Various animal cell types grow during mitosis and cytokinesis. (a) *Left,* schematic of a suspended microchannel resonator (SMR). Single-cell buoyant mass is repeatedly measured as the cell flows back and forth through the vibrating cantilever. *Right,* at cell division, one of the daughter cells is randomly selected and monitored, while the other daughter cell is discarded from the SMR. (b) Buoyant mass trace of a single L1210 cell and its progeny over five full generations. The interdivision time (~9 hr) for cells growing in the SMR and in normal cell culture condition is equivalent. Blue arrows indicate the abscissions of daughter cells. (c) Overlay of 180 individual L1210 cell buoyant mass traces (transparent orange) and the average trace (black) around mitosis. Each mass trace has been normalized so that the typical cell abscission mass is 2. (d) Mass accumulation in mitosis (before

*Figure 1 continued on next page*

*Figure 1 continued*

metaphase/anaphase transition, red) and cytokinesis (blue) relative to the total mass accumulated during the cell cycle for various animal cell types Total relative mass accumulation in M-phase (sum of mitosis and cytokinesis) is indicated on top. Note that while the relative mass accumulation in cytokinesis varies between cell types, all cell types display similar mass accumulation % in early mitosis. n refers to the number of individual cells analyzed. Boxplot line: median, box: interquartile range, whiskers: ± 1.5 x interquartile range.

DOI: https://doi.org/10.7554/eLife.44700.002

The following source data and figure supplements are available for figure 1:

**Source data 1.** L1210 buoyant mass measurement data.
DOI: https://doi.org/10.7554/eLife.44700.005

**Figure supplement 1.** Suspended microchannel resonator (SMR) setups and noise characterization.
DOI: https://doi.org/10.7554/eLife.44700.003

**Figure supplement 2.** Detection of cell cycle transitions.
DOI: https://doi.org/10.7554/eLife.44700.004

The extent of total cell growth during mitosis and cytokinesis was surprising. To determine how generalizable this finding is, we repeated our measurements in other animal cell types. Mouse FL5.12 and BaF3 pro-B lymphocytes, and chicken DT40 lymphoblasts grew 9–13% of their total mass in M-phase (*Figure 1d*). Suspension HeLa (S-HeLa) cells also grew 14% of their total mass during M-phase, validating that substantial growth in M-phase is not specific to lymphocytes. We also examined CD3 +and CD8+activated primary human T cells. Both T-cell subpopulations added approximately 7% of their total mass during M-phase. Thus, growth in mitosis and cytokinesis is an important contributor to the total cellular growth across a variety of cell types grown in suspension.

## Cell mass accumulation persists through prophase, stops as cells approach metaphase-to-anaphase transition and recovers during late cytokinesis

To study the dynamics of cell growth during M-phase, we quantified the absolute mass accumulation rates (MAR) before and during M-phase (*Figure 2—figure supplement 1*; Materials and methods). To account for cell-size-dependent growth rates (*Miettinen and Björklund, 2016*; *Miettinen et al., 2017*; *Son et al., 2012*), we normalized MAR to the mass of the cell (MAR/mass). Surprisingly, after mitotic entry (during approximate prophase) L1210 cells exhibited on average 15.8 ± 3% (mean ± SEM, n = 180) increase in MAR when compared to late G2 phase (*Figure 2a,b*). As cells approached the metaphase-to-anaphase transition, MAR rapidly decreased and eventually reached zero at the end of metaphase. MAR remained near zero for the approximate duration of anaphase, after which MAR started to recover during late cytokinesis (*Figure 2a,c,d*). The recovery of MAR continued through the abscission of daughter cells (*Figure 2d*). These cell growth dynamics also persisted under different nutrient conditions and were reproducible with different SMR devices over multiple years of study (*Figure 2—figure supplement 2a,c*).

We next examined MAR in other cell types. Although increased MAR during early mitosis was only observed in some cell types, MAR remained high after mitotic entry (during approximate prophase) for all the cell types studied (*Figure 2e*; *Figure 2—figure supplement 2b*). All cell types displayed rapid reduction of MAR during metaphase (possibly starting in late prometaphase), near zero MAR during anaphase and a recovery of MAR during late cytokinesis. The temporary stop of cell growth in anaphase was consistently short (<15 min) and coincided with the physical separation of the daughter cells (*Figure 2c–e*). Notably, some cells displayed a negative MAR, indicating a small loss of cell mass during late metaphase and/or early anaphase (*Figure 2e*; *Figure 2—figure supplement 2b*). Together, these results indicate that the mitotic growth behavior is conserved across various animal cell types in suspension, suggesting a role for these specific growth dynamics during mitosis.

## Mitotic protein synthesis rates are consistent with mitotic MAR dynamics

Proteins constitute approximately 70% of cellular dry mass (*Palm and Thompson, 2017*), making it likely that the measured MAR dynamics reflect protein synthesis rates. Using L1210 cells as a model, we quantified the dynamics of mitotic protein synthesis using O-propargyl-puromycin (OPP)-based

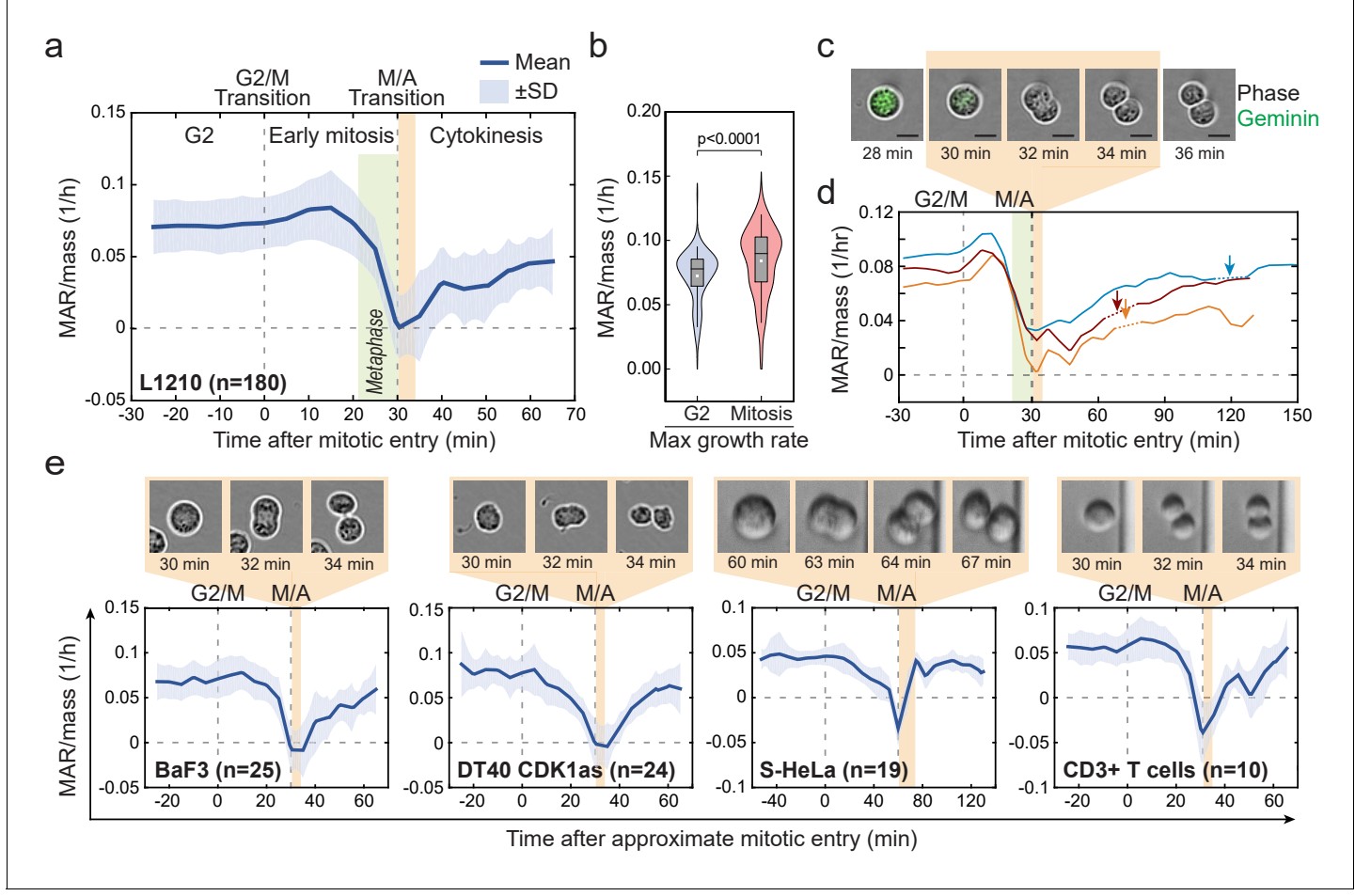

**Figure 2.** Cell mass accumulation persists through prophase, stops as cells approach metaphase-to-anaphase transition and recovers during late cytokinesis. (a) Mass-normalized mass accumulation rate (MAR) of L1210 cells in late G2 and M-phase. G2/M and metaphase-to-anaphase transitions are indicated with dashed vertical lines. Typical durations of metaphase and cell elongation (singlet to doublet) are indicated in green and light brown areas, respectively. n refers to number of individual cells analyzed. (b) Quantification of L1210 cell maximal growth rate in late G2 and in mitosis (n = 180 cells). p-Values obtained using two-tailed Welch's t-test. In boxplots, line: median, box: interquartile range, whiskers: 5–95%. (c) Representative L1210 cell phase contrast (grey) and mAG-hGeminin cell cycle reporter (green) images (n = 18 cells). Times correspond to (a) and (d). Note that the physical separation of daughter cells takes place when cells are not accumulating mass. (d) Examples of individual L1210 mass-normalized MAR traces in late G2, M-phase and early G1. Arrows indicate the final abscission of daughter cells, around which mass-normalized MAR is indicated with dashed lines. M/A denotes the metaphase-to-anaphase transition, G2/M denotes the approximate mitotic entry, both of which are indicated with dashed vertical lines. (e) Mass-normalized MAR of indicated cell types along with representative images displaying the duration of the physical separation of daughter cells. BaF3 and DT40 cells were imaged separately, whereas S-HeLa and CD3 +T cells were imaged on-chip simultaneously with MAR measurements. M/A denotes the metaphase-to-anaphase transition, G2/M denotes the approximate mitotic entry, both of which are indicated with dashed vertical lines. Solid dark blue lines indicate the mean and light blue areas represent ± SD. n refers to number of individual cells analyzed.
DOI: https://doi.org/10.7554/eLife.44700.006

The following figure supplements are available for figure 2:

**Figure supplement 1.** Mass accumulation rate (MAR) analysis.
DOI: https://doi.org/10.7554/eLife.44700.007

**Figure supplement 2.** MAR in different growth conditions and cell types.
DOI: https://doi.org/10.7554/eLife.44700.008

single-cell protein synthesis assays (*Liu et al., 2012*) together with a mitotic marker (phospho-His-tone H3 (Ser10)) (*Figure 3a*; *Figure 3—figure supplement 1a,b*; Materials and methods). For unsynchronized cells, the average mitotic protein synthesis rates were 85 ± 6% (mean ± SD, n = 6) of the rate for G2 cells. We then synchronized cells using double thymidine block followed by CDK1 inhibitor (RO-3306)-mediated G2 arrest and a release, which was followed by 10 min OPP labelling at

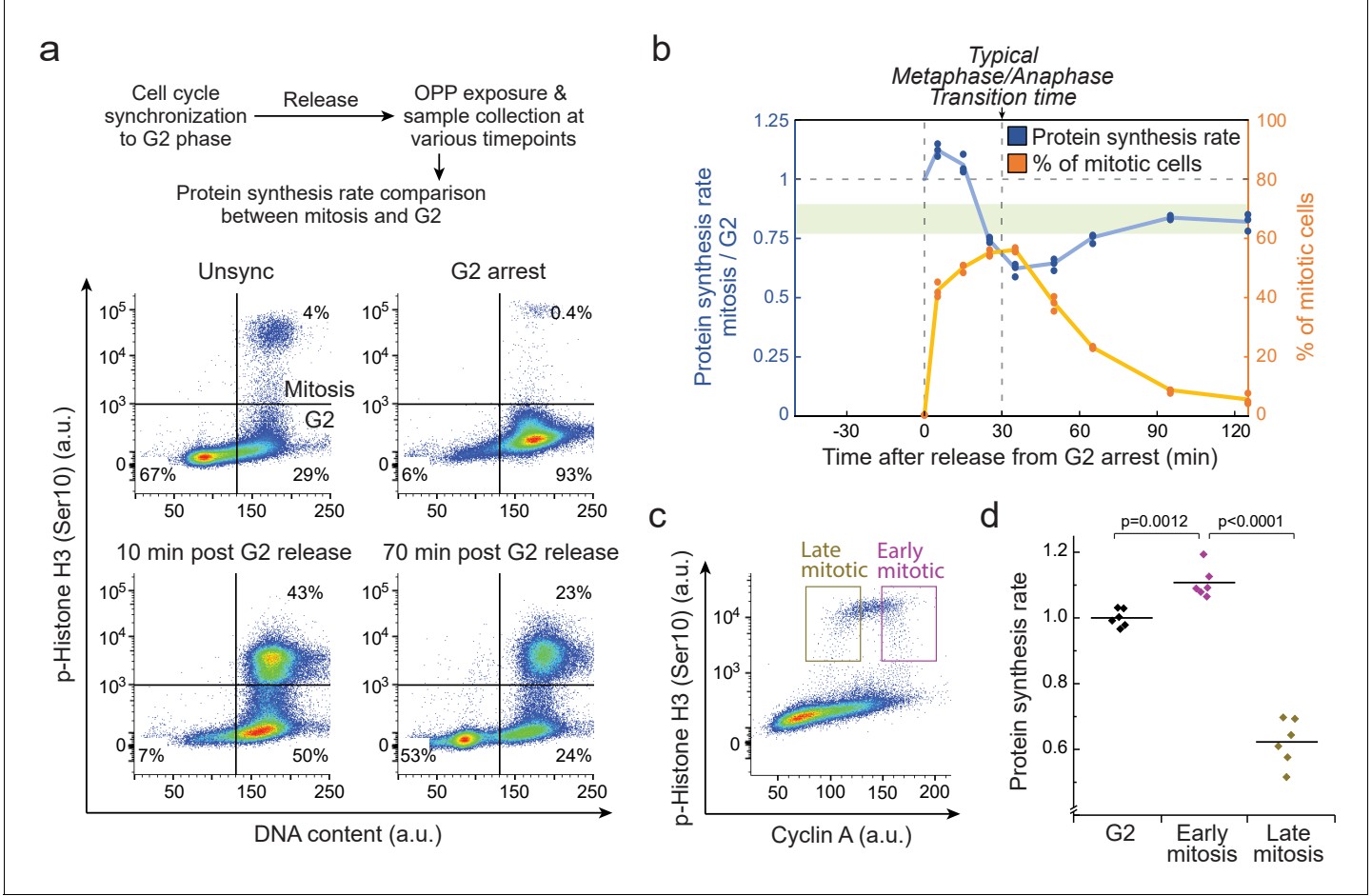

**Figure 3.** Mitotic protein synthesis dynamics are consistent with mass accumulation dynamics. (**a**) *Top*, schematic of the protocol for quantifying mitotic protein synthesis rates using O-propargyl-puromycin (OPP). G2 synchronization was achieved by double thymidine block followed by RO-3306 mediated G2 arrest. *Bottom*, representative FACS scatter plots indicating L2110 cell cycle synchrony (n = 3 independent experiments). Phospho-Histone H3 (Ser10) was used as a mitotic marker. (**b**) Ratio of protein synthesis rate (blue) between mitotic and G2 L1210 cells after release from G2 arrest. Light green area displays the typical protein synthesis ratio between mitotic and G2 cells in the absence of cell cycle synchronization. The relative portion of mitotic cells is shown in orange. Each data point represents an individual replicate. (n = 3 separate cultures for each timepoint). Time of G2 release and the typical time to reach metaphase-to-anaphase transition are indicated with dashed vertical lines. (**c**) Representative FACS scatter plot indicating the separation of early (prophase) and late (metaphase to telophase) mitotic L1210 cells using Cyclin A antibody staining. (**d**) Protein synthesis rate of G2, early mitotic and late mitotic L1210 cells. (n = 6 separate cultures). Early and late mitotic cells were separated as shown in (**g**). p-Values obtained using ANOVA followed by Tukey's posthoc test.

DOI: https://doi.org/10.7554/eLife.44700.009

The following figure supplement is available for figure 3:

**Figure supplement 1.** Single-cell protein synthesis assays in mitotic cells.

DOI: https://doi.org/10.7554/eLife.44700.010

various timepoints. We normalized mitotic protein synthesis rates to G2 protein synthesis rates to avoid cell synchronization induced biases (*Figure 3a*). Immediately after the release from G2 arrest, when mitotic cells were in prophase, protein synthesis rates were higher in mitotic cells than in G2 cells (*Figure 3b*). Approximately 30 min later, when most mitotic cells proceeded to cytokinesis, the protein synthesis rates were reduced to below the normal G2 levels.

To validate that the observed protein synthesis dynamics are not an artefact of cell synchronization, we utilized a previously developed approach where Cyclin A, which is degraded during prometaphase (*den Elzen and Pines, 2001*), is used to separate early (prophase) and late (metaphase to telophase) mitotic cells (*Ly et al., 2017*) (*Figure 3c*). In the absence of cell cycle synchronization,

OPP-based protein synthesis assay indicated that early mitotic cells have higher protein synthesis rates than G2 cells, whereas in late mitotic cells, protein synthesis is reduced (*Figure 3d*). We also separated G2, early mitosis and late mitosis based on cyclin B1, which is degraded at metaphase-to-anaphase checkpoint (*Figure 3—figure supplement 1a*). This approach also revealed that protein synthesis rates remain high in early mitosis but not in late mitosis (*Figure 3—figure supplement 1c*). Although protein synthesis assays do not have temporal resolution required for separation of all mitotic stages, the protein synthesis dynamics we observe correspond to those observed with MAR measurements. In conclusion, L1210 cells display increased growth in early mitosis and radically reduced growth in metaphase and anaphase.

## Mitotic arrests, including antimitotic chemotherapies, inhibit cell growth

Our results show that growth is inhibited from metaphase (or late prometaphase) to the end of anaphase (*Figure 2a*). As many studies examine mitotic growth by arresting cells to mitosis, and this has been suggested to reduce cell growth (*Coldwell et al., 2013*; *Sivan et al., 2011*; *Stonyte et al., 2018*), we measured the effect of chemically induced mitotic arrest on cell growth. First, we monitored the MAR of L1210 cells treated with kinesin inhibitor S-trityl-l-cysteine (STLC), which arrests cells in prometaphase state. These cells displayed a growth burst in early mitosis, similarly to untreated cells, after which MAR approached zero over the course of 2–3 hr (*Figure 4a*). We also repeated our mitotic protein synthesis assay (*Figure 3a*) in the presence of STLC. Similarly to MAR, protein synthesis rates increased after mitotic entry, but then gradually decreased as cells were arrested in mitosis (*Figure 4b*).

To separate drug-specific effects on growth from those that reflect mitotic arrest, we tested three additional chemical approaches for arresting cells in mitosis. These were microtubule inhibitor nocodazole, proteasome inhibitor MG-132 and Anaphase-Promoting Complex inhibitor proTAME (*Zeng et al., 2010*). All these chemicals resulted in similar reduction in the overall mitotic protein synthesis (*Figure 4c*). None of these chemicals caused protein synthesis to be significantly reduced in G2, except for MG-132. In addition, nocodazole treatment resulted in identical MAR behavior as STLC (*Figure 4a*).

Mitotic arrest is the mechanism of action for many chemotherapy drugs. We examined how clinically relevant concentrations of the chemotherapy drugs Vinblastine and Vincristine (*Florian and Mitchison, 2016*) affect cell MAR. Neither of the drugs affected cell growth in G2, but as cells were arrested in mitosis, their growth rate reduced to zero (*Figure 4d*). Thus, mitotic arrests, including antimitotic chemotherapies, stop cell growth independently of the arresting mechanism.

## Cells in metaphase and anaphase display mitotic stage specific inhibition of mass accumulation

Next, we studied if metaphase and anaphase, where MAR was near zero, have a growth reducing mechanism(s) that is not active earlier in mitosis. We first considered the role of mitotic deswelling. During mitosis cells round up and increase their volume by approximately 10–20%, before shrinking (deswelling) back to their original volume during anaphase (*Son et al., 2015a*; *Zlotek-Zlotkiewicz et al., 2015*). While mitotic rounding has minimal influence on the cell types used in this study, as the suspension cells display a spherical morphology throughout the cell cycle, inhibition of mitotic cell swelling removed the MAR increase seen in early mitosis but did not affect MAR in metaphase and anaphase (*Figure 5—figure supplement 1a–c*). Furthermore, inhibition of mitotic swelling did not influence early mitotic protein synthesis rates (*Figure 5—figure supplement 1d*). Thus, while mitotic swelling influences the MAR observed in early mitosis, possibly by increasing cell mass due to the uptake of ions, this does not explain why cells suddenly stop growing around metaphase-to-anaphase transition.

Next, we examined if cytokinetic cell elongation is required for the near zero (or even negative) MAR around metaphase-to-anaphase transition. We treated L1210 cells with Tozasertib, an Aurora kinase inhibitor, which blocks cytokinesis but not mitosis as evident from the loss of Geminin (*Figure 5a*). Tozasertib-treated cells displayed low MAR around metaphase-to-anaphase transition, although MAR remained higher than in control cells (*Figure 5b*; *Figure 5—figure supplement 2a*). We then treated cells with blebbistatin, a myosin motor inhibitor which also blocks cytokinesis

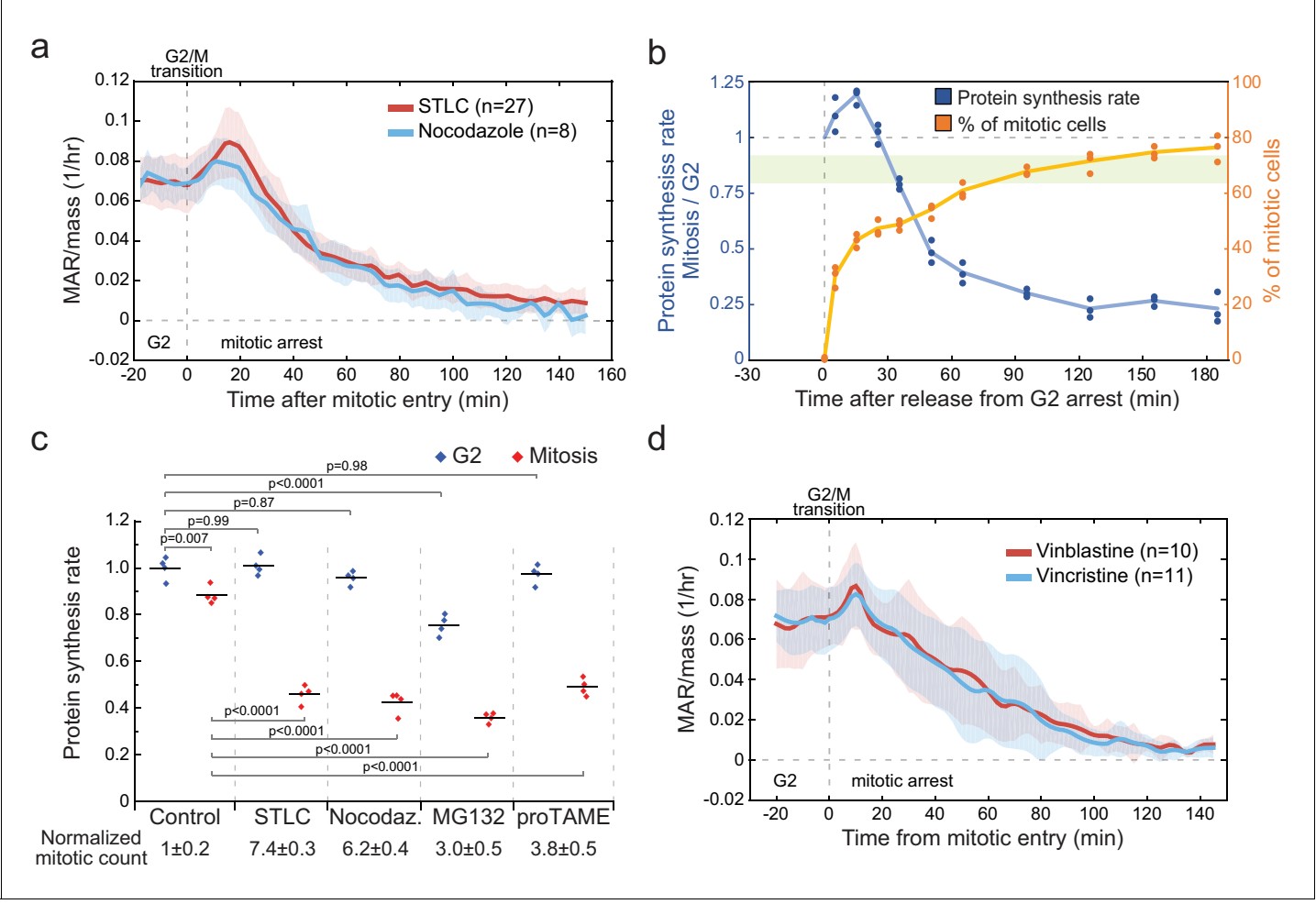

**Figure 4.** Mitotic arrests result in growth inhibition independently of the mechanism of arrest. (a) Mass-normalized MAR of 5 µM STLC or 1 mg/ml Nocodazole treated L1210 cells in late G2 and mitosis. Dashed vertical line indicates the approximate mitotic entry. Solid dark lines indicate the mean and light areas represent ± SD. n refers to number of individual cells analyzed. Drug treatments started 1–4 hr prior to mitotic entry and were maintained through the experiment. (b) Ratio of protein synthesis rate (blue) between mitotic and G2 L1210 cells after release from G2 arrest in to 5-µM STLC-mediated mitotic arrest. Light green area displays the typical protein synthesis ratio between mitotic and G2 cells in the absence of cell cycle synchronization. The relative portion of mitotic cells is shown in orange. Each data point represents an individual replicate (n = 3 separate cultures for each timepoint). Cells were synchronized to G2 as in (*Figure 3a*). (c) Protein synthesis rates of G2 (blue) and mitotic (red) L1210 cells after 4 hr treatment with indicated mitotic inhibitors. The proportion of mitotic cells relative to control is indicated below (mean ± SD). (n = 4 separate cultures). p-Values obtained using ANOVA followed by Tukey's posthoc test. (d) Mass-normalized MAR of 10 nM Vinblastine or 10 nM Vincristine-treated L1210 cells in late G2 and mitosis. Dashed vertical line indicates the approximate mitotic entry. Solid dark lines indicate the mean and light areas represent ± SD. n refers to number of individual cells analyzed. Drug treatments started 1–4 hr prior to mitotic entry and were maintained through the experiment.

DOI: https://doi.org/10.7554/eLife.44700.011

(*Atilla-Gokcumen et al., 2010*). Blebbistatin treatment resulted in MAR dynamics comparable to control cells (*Figure 5—figure supplement 2b*). In addition, both Tozasertib and blebbistatin prolonged the duration of early mitosis. Together, these data indicate that the physical separation of daughter cells does not explain the observed MAR dynamics.

The radical reduction in MAR observed as cells approach metaphase-to-anaphase transition could be explained by two separate mechanisms: First, growth may be reduced as a function of time after mitotic entry, or possibly after an initial delay in growth reduction. This hypothesis is supported by the gradual decrease in growth rates following mitotic arrests (*Figure 4*), possibly because of the inhibition of transcription as DNA condenses. Second, there may be a separate growth inhibiting mechanism(s) specific to metaphase and anaphase. To separate these two options, we compared the MAR as a function of time from mitotic entry in control cells and cells arrested in prometaphase

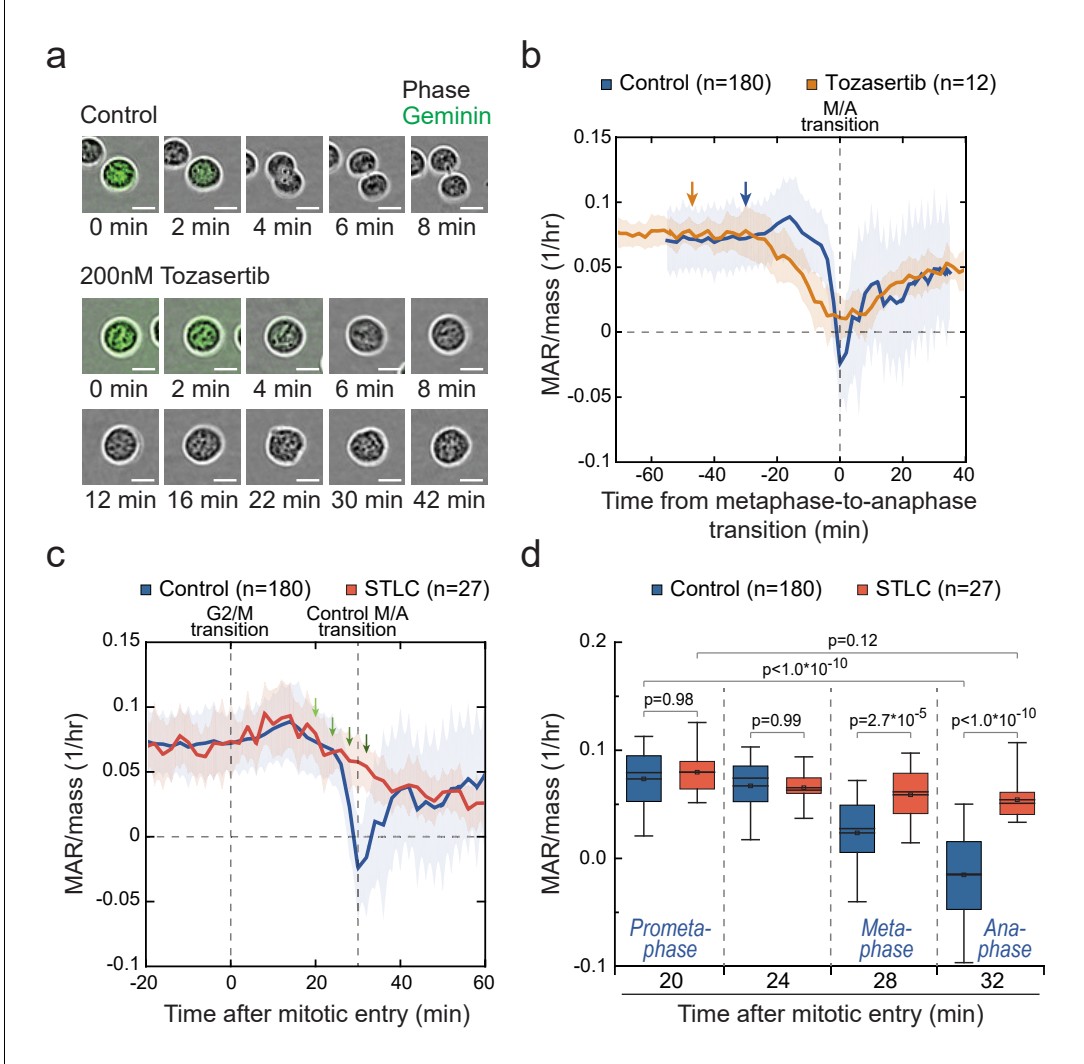

**Figure 5.** Cells in metaphase and anaphase display stage-specific mass accumulation regulation independently of cell elongation. (**a**) Representative L1210 cell phase contrast (grey) and mAG-hGeminin cell cycle reporter (green) images in control (n = 9 cells) and 200 nM Tozasertib (n = 6 cells) treated cells. The degradation of mAG-hGeminin indicates metaphase-to-anaphase transition. No cytokinesis takes place under Tozasertib treatment. (**b**) Mass-normalized MAR of control (blue) and 200 nM Tozasertib (orange) treated L1210 cells. Note that Tozasertib prolonged early mitosis, but most cells still displayed increased MAR after G2/M transition (see *Figure 5—figure supplement 2a*). Dashed vertical line indicates the metaphase-to-anaphase transition. Solid dark lines indicate the mean and light areas represent ± SD. n refers to number of individual cells analyzed. Arrows reflect typical time of G2/M transition for each sample. Drug treatment started 1–4 hr prior to mitotic entry and was maintained through the experiment. (**c, d**) Mass-normalized MAR of control (blue) and 5 µM STLC (red) treated L1210 cells (**f**). Dashed vertical lines indicate the approximate mitotic entry (for both samples) and metaphase-to-anaphase transition (only applies to control). Solid dark lines indicate the mean and light areas represent ± SD. Arrows indicate time points from which data was extracted to generate the boxplot in (**g**). n refers to number of individual cells analyzed. p-Values were obtained using ANOVA followed by Tukey's posthoc test. In all boxplots, line: median, box: interquartile range, whiskers: 5–95%. See Materials and methods for details on MAR analysis resolution for this figure.

DOI: https://doi.org/10.7554/eLife.44700.012

The following figure supplements are available for figure 5:

**Figure supplement 1.** Mitotic cell swelling affects MAR, but not protein synthesis in early mitosis.
DOI: https://doi.org/10.7554/eLife.44700.013

**Figure supplement 2.** The low MAR in metaphase and anaphase are not explained by cell elongation or time spent in mitosis.
DOI: https://doi.org/10.7554/eLife.44700.014

state using STLC (*Figure 5c*). 20 min after mitotic entry, when cells are in late prometaphase, both samples displayed similar growth rates (*Figure 5d*). However, 28 min and 32 min after mitotic entry, when control cells had proceeded to metaphase and anaphase, respectively, but STLC-treated cells remained arrested in prometaphase, the control cells displayed lower MAR. Similar results were obtained when prometaphase arrest was achieved using Nocodazole (*Figure 5—figure supplement 2c*). In conclusion, the mitotic morphological changes (*Figure 5a,b*; *Figure 5—figure supplement 1*; *Figure 5—figure supplement 2a,b*) and the time that cells have spent in mitosis cannot fully explain the observed MAR in metaphase and anaphase. Thus, additional MAR reducing mechanism(s) must exist around metaphase-to-anaphase transition.

## Mitotic growth does not require mTOR activity

We then investigated what signaling promotes growth and protein synthesis in mitosis (*Figure 6a*). We measured the levels of phosphorylated 4E-BP1 (Thr37/46) and S6RP (Ser235/236) at the single-cell level in mitotic and G2 L1210 cells. Mechanistic Target Of Rapamycin (mTOR) regulates both of these proteins, which in turn control translation (*Fingar et al., 2004*). Importantly, 4E-BP1 is a negative regulator of translation and is inactivated by phosphorylation on several sites, including Thr37/46 (reviewed by *Qin et al., 2016*). Phosphorylated 4E-BP1 levels were approximately threefold higher in mitosis than in G2 (*Figure 6b*; *Figure 6—figure supplement 1*), whereas phosphorylated S6RP displayed only a minor increase in mitosis (*Figure 6—figure supplement 2a*). The mitotic phosphorylation of 4E-BP1 was validated with an independent antibody using microscopy and the mitotic increase was not observed when using antibody isotype controls or pretreating the sample using Lambda protein phosphatase (*Figure 6—figure supplement 1*). The mitotic phosphorylation of 4E-BP1 is also consistent with previous reports (*Shuda et al., 2015*) and the observation that the translational targets of mTOR are actively translated during mitosis (*Park et al., 2016*).

To examine the role of mTOR, we treated cells for 2 hr with 250 nM mTOR inhibitor TORIN-1. In G2 cells, the levels of phosphorylated 4E-BP1 (Thr37/46) and S6RP (Ser235/236) were near zero (*Figure 6b*; *Figure 6—figure supplement 2b*). In mitosis, phosphorylation of S6RP was reduced by TORIN-1 (*Figure 6—figure supplement 2b*), indicating that mTOR remains active in mitosis. However, TORIN-1 did not change the levels of phosphorylated 4E-BP1 in mitosis (*Figure 6b*), suggesting that an mTOR-independent mechanism activates 4E-BP1 in mitosis. Next, we measured mitotic protein synthesis rates in the presence of TORIN-1. Although G2 protein synthesis rates were reduced, mitotic protein synthesis rates were not affected by TORIN-1 (*Figure 6—figure supplement 2c*). Thus, mTOR is not a major contributor to mitotic growth.

## CDK1 drives phosphorylation of 4E-BP1, protein synthesis and mass accumulation in mitosis

We examined how the levels of phosphorylated 4E-BP1 (Thr37/46) and S6RP (Ser235/236) change when CDK1 is inhibited with 1 μM RO-3306 (*Vassilev et al., 2006*). At this RO-3306 concentration, L1210 cells can still progress through mitosis, as CDK1 is only partially inhibited (*Son et al., 2015a*). Only in mitosis did RO-3306 reduce phosphorylated 4E-BP1 but not phosphorylated S6RP (*Figure 6b*; *Figure 6—figure supplement 2b*). Although the role of CDK1 in controlling 4E-BP1 phosphorylation has been reported before (*Heesom et al., 2001*; *Shuda et al., 2015*), the consequences of this on protein synthesis and cell growth remain unknown. We observed that 1 μM RO-3306 treatment also reduced mitotic protein synthesis rates without affecting G2 protein synthesis rates (*Figure 6c*). In addition, RO-3306 treatment prolonged early mitosis and resulted in a clear reduction of MAR in mitosis but not in G2 (*Figure 6d*).

To further validate the role of CDK1 in promoting mitotic growth, we utilized chemical genetics in the DT40 CDK1as cell line. In these cells, the wild-type CDK1 has been replaced with *Xenopus laevis* CDK1 that has a F80G mutation (*Gibcus et al., 2018*), which sensitizes CDK1 to inhibition by the ATP analog 1NM-PP1 (*Hochegger et al., 2007*). We observed that protein synthesis was reduced by 1NM-PP1 in a dose-dependent manner in mitosis but not in G2 (*Figure 6e*). Consistently, 1NM-PP1 also reduced MAR in mitosis (*Figure 6f*).

We also investigated the role of other mitotic kinases in promoting protein synthesis and cell growth. Inhibitors for CDK2, Aurora kinases and DYRK kinases did not affect mitotic protein synthesis (*Figure 6—figure supplement 3a*). OTSSP167, a drug designated as a MELK inhibitor

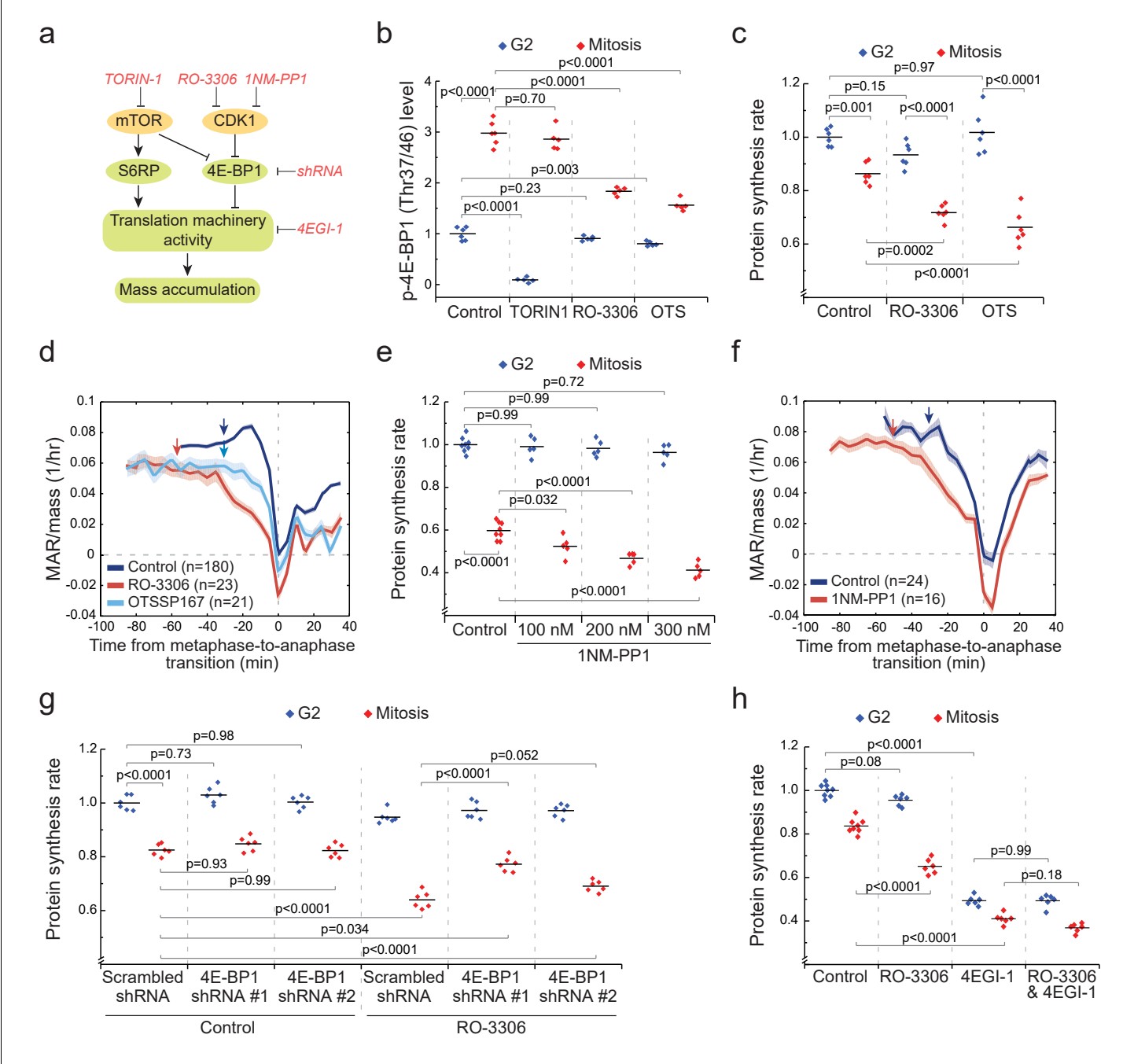

**Figure 6.** CDK1 drives mitotic growth through 4E-BP1 and cap-dependent protein synthesis. (a) Schematic of growth regulation pathways. Chemical and genetic inhibitors (red), kinases (yellow) and the measured downstream consequences (green) are shown. 1NM-PP1-mediated inhibition of CDK1 is dependent on kinase mutation. (b) L1210 cell levels of phosphorylated 4E-BP1 (Thr37/46) in G2 (blue) and mitosis (red) after 2 hr treatment with 250 nM TORIN-1, 1 μM RO-3306 or 50 nM OTSSP167. (n = 5–6 separate cultures). (c) Protein synthesis rates of G2 (blue) and mitotic (red) L1210 cells after 2 hr treatment with 1 μM RO-3306 or 50 nM OTSSP167. (n = 6 separate cultures). (d) Mass-normalized MAR of control, 1 μM RO-3306 or 30 nM OTSSP167-treated L1210 cells. Solid dark lines indicate the mean and light areas represent ± SEM. Arrows reflect typical time of G2/M transition for each sample. n refers to number of individual cells analyzed. Drug treatments started 1–4 hr prior to mitotic entry and were maintained through the experiment. (e) DT40 CDK1as cell protein synthesis rates in G2 (blue) and mitosis (red) after 5 hr treatment with 1NM-PP1. (n = 5–8 separate cultures). (f) Mass-normalized MAR of control or 200 nM 1NM-PP1-treated DT40 CDK1as cells. Solid dark lines indicate the mean and light areas represent ± SEM. Arrows reflect typical time of G2/M transition for each sample. n refers to number of individual cells analyzed. Drug treatments started 1–4 hr prior to mitotic entry and were maintained through the experiment. (g) Protein synthesis rates of G2 (blue) and mitotic (red) L1210 cells expressing scrambled or 4E-BP1 targeting shRNAs. The cells were treated for 2 hr with 1 μM RO-3306 before sample preparation. (n = 6 separate cultures). (h) Protein synthesis rates of

*Figure 6 continued on next page*

*Figure 6 continued*

G2 (blue) and mitotic (red) L1210 cells after 2 hr treatment with 1 µM RO-3306, 5 hr treatment with 50 µM 4EGI-1 or combined treatment with RO-3306 and 4EGI-1. (n = 6–8 separate cultures). All p-Values were obtained using ANOVA followed by Tukey's posthoc test.

DOI: https://doi.org/10.7554/eLife.44700.015

The following figure supplements are available for figure 6:

**Figure supplement 1.** 4E-BP1 is phosphorylated in mitosis.
DOI: https://doi.org/10.7554/eLife.44700.016

**Figure supplement 2.** mTOR is active in mitosis but is not required for mitotic protein synthesis.
DOI: https://doi.org/10.7554/eLife.44700.017

**Figure supplement 3.** Kinase inhibitors which have MELK kinase as an off-target do not reduce mitotic protein synthesis.
DOI: https://doi.org/10.7554/eLife.44700.018

**Figure supplement 4.** shRNA-mediated knockdown of 4E-BP1 in L1210 cells.
DOI: https://doi.org/10.7554/eLife.44700.019

(*Chung et al., 2012*), reduced phosphorylation of 4E-BP1, protein synthesis and mass accumulation rates in mitosis (*Figure 6b–d*). Consistently, MELK has been suggested to promote translation in mitosis (*Wang et al., 2016*). However, alternative MELK inhibitors (*Klaeger et al., 2017*) did not display mitosis-specific inhibition of protein synthesis (*Figure 6—figure supplement 3b,c*), suggesting that the mitotic growth effects observed in OTSSP167-treated cells were not mediated by MELK but by OTSSP167 off-targets.

We then moved to validate that 4E-BP1 mediates the CDK1 driven protein synthesis. We generated two L1210 cell lines expressing 4E-BP1 targeting shRNAs, which reduced 4E-BP1 levels by approximately 85% (shRNA #1) and by 50% (shRNA #2) (*Figure 6—figure supplement 4a,b*). The 4E-BP1 knockdowns had little effect on proliferation rate (*Figure 6—figure supplement 4c*) or G2 cell protein synthesis (*Figure 6g*), possibly reflecting that 4E-BP1 is kept mostly inactive under our experimental conditions. However, when we reduced mitotic protein synthesis by inhibiting CDK1 with RO-3306, the knockdowns of 4E-BP1 partially rescued the mitotic protein synthesis inhibition (*Figure 6g*). These data indicate that CDK1 promotes mitotic growth at least partly through 4E-BP1.

4E-BP1-driven cap-dependent protein synthesis is classically considered to be inhibited in mitosis (*Bonneau and Sonenberg, 1987*; *Pyronnet et al., 2001*), although recently this view has been challenged (*Coldwell et al., 2013*; *Shuda et al., 2015*). We therefore tested if i) cap-dependent translation remains active in L1210 cell mitosis, and if ii) cap-dependent translation is required for CDK1-mediated mitotic growth. To test these, we first inhibited cap-dependent translation using 4EGI-1, an inhibitor of eIF4F complex assembly (*Cencic et al., 2011*). Both G2 and mitotic protein synthesis rates were reduced by a similar amount, approximately 50%, following 4EGI-1 treatment (*Figure 6h*). However, treatment with both 4EGI-1 and RO-3306 did not significantly change mitotic protein synthesis rates when compared to treatment with 4EGI-1 alone, suggesting that cap-dependent protein synthesis is involved in CDK1-driven mitotic translation. In conclusion, our results are consistent with a previous report (*Shuda et al., 2015*) that CDK1 substitutes for mTOR in mitosis to promote phosphorylation of 4E-BP1, to maintain cap-dependent translation and to promote mass accumulation.

## CDK1-driven mitotic protein synthesis supports daughter cell growth

Mitotic transcription and translation have been suggested to be geared toward ribosomal proteins and other components that promote growth (*Aviner et al., 2017*; *Palozola et al., 2017*). Therefore, inhibiting mitotic protein synthesis could also impact growth in cytokinesis and in daughter cells. We compared the MAR of RO-3306 or OTSSP167-treated L1210 cells to control cells in late G2 (last 30 min before G2/M transition), early mitosis (before metaphase-to-anaphase transition), cytokinesis (after metaphase-to-anaphase transition) and newborn G1 (first 30 min after abscission of daughter cells). In the presence of the mitotic growth inhibitors, G2 MAR was not affected, but MAR remained low in cytokinesis and in newborn G1 (*Figure 7a*). In contrast, cells treated with 100 nM cycloheximide, a translation inhibitor which at this concentration reduces total protein synthesis by approximately 50% (*Figure 3—figure supplement 1d*), did not display similar cell cycle specificity in growth inhibition (*Figure 7a*). In addition, mother cell MAR in early mitosis and daughter cell MAR in early G1 correlated in both control ($R^2 = 0.42$) and RO-3306 ($R^2 = 0.33$) treated cells (*Figure 7—figure*

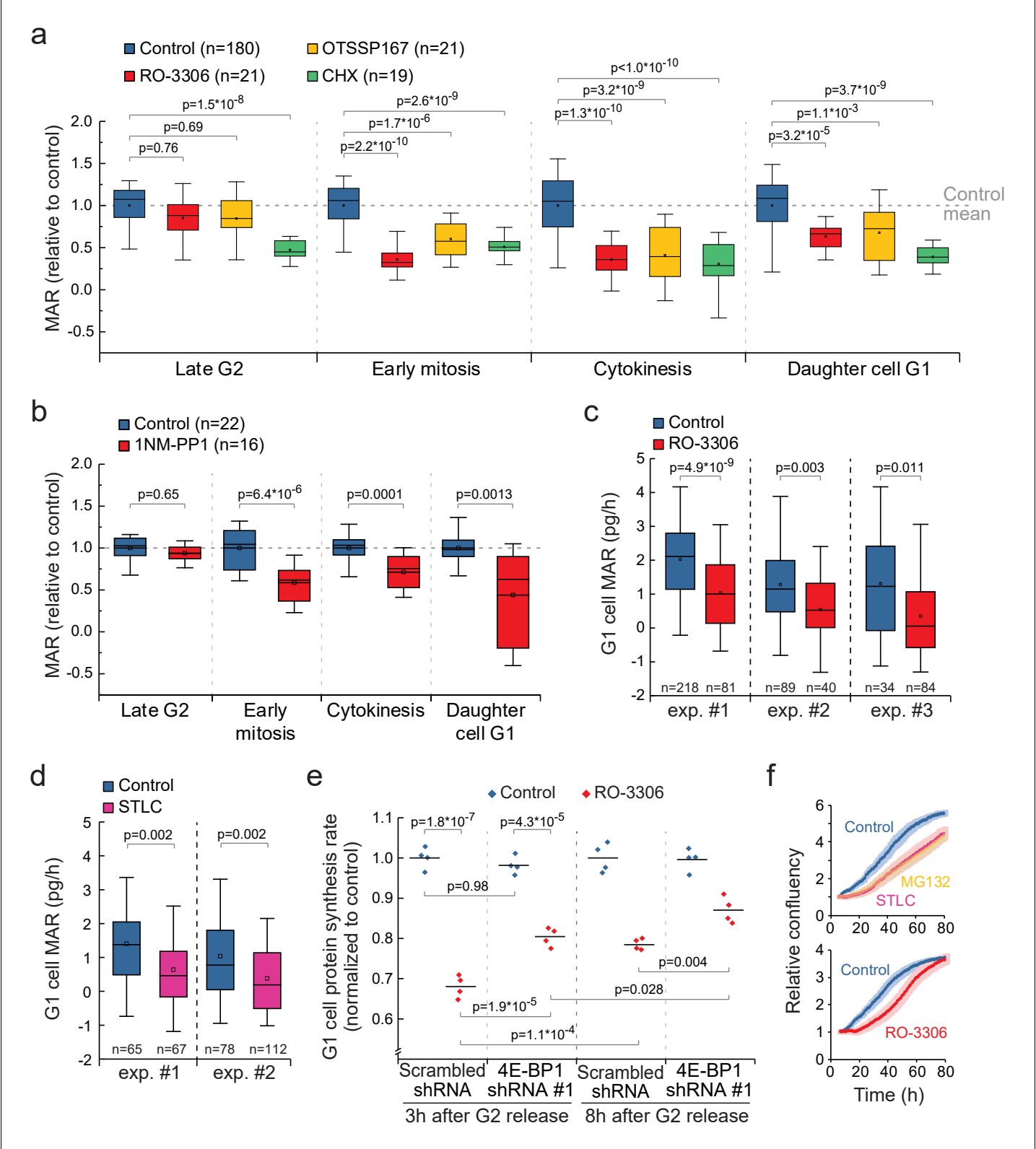

**Figure 7.** CDK1-driven mitotic protein synthesis supports daughter cell growth. (**a**) MAR during indicated stages of cell cycle in control, 1 µM RO-3306, 30 nM OTSSP167 or 100 nM cycloheximide (CHX) treated L1210 cells. The MAR values were normalized to control mean at each stage. n refers to number of individual cells analyzed. Drug treatments started 1–4 hr prior to mitotic entry and were maintained through the experiment. (**b**) MAR during indicated stages of cell cycle in control or 200 nM 1NM-PP1-treated DT40 CDK1as cells. The MAR values were normalized to control mean at each

*Figure 7 continued on next page*

*Figure 7 continued*

stage. n refers to number of individual cells analyzed. Drug treatment started 1–4 hr prior to mitotic entry and was maintained through the experiment. (**c** and **d**) MAR of newborn L1210 G1 cells from control and from cells that have undergone mitosis in the presence of 1 µM RO-3306 (**c**) or from cells that have been arrested to mitosis for 4 hr with STLC before releasing to undergo cytokinesis (**d**). Data acquired using serial SMR (see ***Figure 7—figure supplement 2*** for details). n refers to number of individual cells analyzed. (**e**) Protein synthesis rates of G1 L1210 cells expressing a scrambled or 4E-BP1 targeting shRNA after the cells progressed through mitosis in the presence or absence of 1 µM RO-3306. Two timepoints (3 hr and 8 hr) after G2 release are shown. (n = 4 separate cultures). (**f**) Long-term growth, as measured by cell confluency, in L1210 cells that have been arrested to mitosis for 4 hr with STLC or MG132 before releasing to undergo cytokinesis (n = 5 separate cultures, top), or have undergone mitosis in the presence of 1 µM RO-3306 (n = 6 separate cultures, bottom). Dark colors indicate mean and light areas indicate ± SEM. In (**a**) and (**e**), p-values obtained using ANOVA followed by Tukey's posthoc test. In (**b–d**), p-values obtained using two-tailed Welch's t-test. In boxplots, line: median, box: interquartile range, whiskers: 5–95%.

DOI: https://doi.org/10.7554/eLife.44700.020

The following figure supplements are available for figure 7:

**Figure supplement 1.** Correlations between mitotic and G1 cell MAR.

DOI: https://doi.org/10.7554/eLife.44700.021

**Figure supplement 2.** Daughter cell growth rate measurement workflow.

DOI: https://doi.org/10.7554/eLife.44700.022

**Figure supplement 3.** Daughter cells protein synthesis rates in G1 following mitotic growth inhibitions.

DOI: https://doi.org/10.7554/eLife.44700.023

*supplement 1*). 1NM-PP1-mediated inhibition of CDK1 in the DT40 CDK1as cell line also resulted in reduced growth during cytokinesis and in daughter cells (***Figure 7b***).

We speculated that the daughter cell growth inhibition is a consequence of growth reduction in mother cell mitosis. To validate that mitotic growth is needed to promote daughter cell growth, we synchronized L1210 cells to G2 phase, released them back to cell cycle in the presence or absence of mitotic growth inhibitions, and carried out daughter cell growth measurements (see ***Figure 7—figure supplement 2a*** for workflow). First, we measured single-cell MAR using a serial SMR, which measures MAR over a 15–30 min window (***Calistri et al., 2018***; ***Cermak et al., 2016***) (***Figure 7—figure supplement 2b***). We identified the newborn G1 cells based on their smaller mass and quantified their MAR (***Figure 7—figure supplement 2c***). In addition, we quantified protein synthesis rates of G1 daughter cells, as identified through DNA staining, and also measured long-term proliferation of the cells. Following mitotic growth inhibitions, either by temporary mitotic arrest (4 hr STLC treatment) or by partial CDK1 inhibition, newborn cells displayed significantly reduced MAR and protein synthesis rates (***Figure 7c,d***; ***Figure 7—figure supplement 3***). We then repeated the daughter cell protein synthesis assays in 4E-BP1 knockdown cells after the mother cells had progressed through mitosis in the presence or absence of CDK1 inhibition. 4E-BP1 knockdown partly rescued the daughter cell protein synthesis rates that were reduced by CDK1 inhibition (***Figure 7e***). Thus, 4E-BP1 mediates mitotic protein synthesis of CDK1 (***Figure 6***), and this may also affect daughter cell protein synthesis. However, we cannot fully exclude the possibility that daughter cell growth is affected by some other effects of partial CDK1 inhibition, such as chromosome missegregation, which could consequently reduce daughter cell growth independently of mitotic growth. The protein synthesis rate and long-term proliferation rate of the daughter cells recovered over time (***Figure 7e,f***), suggesting that mitotic growth inhibitions do not permanently affect cell viability.

## Discussion

We show that animal cells grow approximately 10% of their total mass during M-phase (***Figure 1***), indicating that the growth during M-phase is comparable to interphase when the short duration of M-phase is taken into consideration. This contradicts the classical dogma that growth takes place primarily during interphase and not during M-phase. Our single-cell measurements show that mass accumulation behavior during mitosis is dynamic (***Figure 2***), dependent on mitotic stage (***Figure 5***), conserved across a variety of animal cell types grown in suspension (***Figure 2***) and reflected in protein synthesis rates (***Figure 3***). Importantly, growth is only stopped for a short duration in mitosis, as cells approach the metaphase-to-anaphase transition, and growth recovers in late cytokinesis after anaphase. Most studies on translation and growth signaling during mitosis use cell-population-based

experimental approaches that require population enrichment for mitotic cells. This is commonly achieved by mitotic blockades, which result in growth inhibition (*Figure 4*). Alternatively, mitotic cells can be collected with mitotic shake-off. However, this only enriches for the small subset of mitotic cells that are temporarily not growing (i.e. metaphase and anaphase cells). Thus, many of the controversial reports regarding translational control during mitosis can be explained by experimental approaches.

Conceptually, the tight and conserved coordination between growth rates and mitotic stages suggests that mitotic growth control is important for cell division. It has been thought that prioritization of energy away from the ATP consuming macromolecule synthesis and towards mechanical reorganization of the cell could explain the reduction in mitotic protein synthesis rates (*Kronja and Orr-Weaver, 2011*; *Sivan and Elroy-Stein, 2008*; *White-Gilbertson et al., 2009*). Consistent with this hypothesis, we show that growth is stopped when the physical separation of daughter cells takes place (*Figure 2*). Yet, there is no direct evidence that cell division would increase the energetic needs of a cell to a point where growth could not persist, and the inhibition of growth may therefore be due to other reasons. For example, the inhibition of growth may protect cells from harmfully excessive cell growth during prolonged mitosis (*Miettinen and Björklund, 2016*; *Miettinen et al., 2017*; *Neurohr et al., 2019*).

It has also been proposed that growth, and especially protein synthesis, are required during mitosis. A complete growth inhibition for the duration of M-phase would result in loss of short-lived proteins needed for mitosis and cytokinesis (*White-Gilbertson et al., 2009*). Consistently, we observe that growth is not inhibited during prophase or late cytokinesis, suggesting that short-lived proteins, such as survivin (*White-Gilbertson et al., 2009*), can be produced immediately prior to and after anaphase. Future studies examining cellular energy production and mapping the levels of short-lived proteins in different mitotic stages will help elucidate why cells display such a dynamic growth behavior in mitosis.

Mechanistically, we found that CDK1 promotes mitotic mass accumulation and protein synthesis, and this is at least partly mediated by 4E-BP1 (*Figure 6*). Our results are consistent with previously reported interactions between CDK1 and the translational machinery (*Heesom et al., 2001*; *Shuda et al., 2015*; *Smith and Proud, 2008*). Importantly, our results do not exclude the existence of other CDK1 dependent or independent mechanisms regulating mitotic growth. Indeed, CDK1 remains active until metaphase-to-anaphase transition (*Gavet and Pines, 2010*), whereas mass accumulation rates begin to decay in late prometaphase and metaphase, indicating that CDK1 alone cannot fully explain the observed mass accumulation dynamics.

Mitotic growth may be dependent on the time that cells have spent in mitosis, as growth can be limited by the reduced chromatin accessibility and consequently reduced transcription. This is consistent with the gradual growth reduction we observe during mitotic arrests (*Figure 4*). However, considering the long half-life of most mRNAs (*Schwanhäusser et al., 2011*), and the observations that mRNA levels do not decline during mitosis (*Novais-Cruz et al., 2018*; *Tanenbaum et al., 2015*), limited chromatin accessibility does not explain why protein synthesis would be reduced in the absence of mitotic arrests. Furthermore, our results also show that normal mitotic progression results in more radical reduction in mass accumulation than what is observed in prometaphase arrested cells (*Figure 5*), indicating that there are also other mechanism(s) that radically reduce mass accumulation around metaphase-to-anaphase transition.

Cell mass accumulation reflects the flux of components, such as nutrients, in and out of the cell. Because we observe that mass accumulation fully stops during anaphase, or even becomes negative in some cell types, the mechanism(s) controlling mass accumulation around metaphase-to-anaphase transition must either block the cells from taking up nutrients or even expel cellular components. Yet, this does not necessarily mean that protein synthesis comes to a complete stop, as cells can maintain translation even in the absence of nutrient uptake by degrading proteins and recycling the components (*Son et al., 2015b*). In fact, prolonged mitotic arrests resulted in near zero mass accumulation although ~25% of the protein synthesis rate persisted (*Figure 4*). We therefore hypothesize that around metaphase-to-anaphase transition, where the anaphase-promoting complex drives protein degradation, cells largely stop nutrient uptake but maintain low level of protein synthesis by recycling amino acids.

We also observed that reducing CDK1 activity results in reduced growth in daughter cells (*Figure 7*). This may be explained by CDK1-driven translation of growth components during mitosis,

such as ribosomal proteins, that are required to 'jump-start' growth in the newborn cells. Consistently, several studies have suggested that transcription and translation of growth-related genes are prioritized during mitosis (*Aviner et al., 2017*; *Palozola et al., 2017*; *Park et al., 2016*). These results imply that CDK1 does not only coordinate cell division, but also optimizes mitotic translation to promote immediate daughter cell growth. However, it should be noted that the active mitotic translation of growth promoting components remains controversial (*Stumpf et al., 2013*; *Tanenbaum et al., 2015*), CDK1 inhibition may reduce daughter cell growth independently of mitotic growth, and the physiological significance of mitotic translational control in vivo remains unexplored.

Finally, our observations that mitotic arrests block cell growth have implications for antimitotic cancer chemotherapy. We observed that the vinca alkaloids Vincristine and Vinblastine, both commonly used to treat lymphomas, stop cell growth when used in concentrations lower than those measured in patient plasma (*Florian and Mitchison, 2016*). This mitotic growth arrest may contribute to the induction of mitotic catastrophe and the efficacy of antimitotic chemotherapies.

## Materials and methods

**Key resources table**

| Reagent type (species) or resource | Designation | Source or reference | Identifiers | Additional information |
|---|---|---|---|---|
| Cell line (*M. musculus*) | L1210 | ATCC | Cat#CCL-219; RRID:CVCL_0382 | |
| Cell line (*M. musculus*) | L1210 - FUCCI | Other | | Generated in a previous study (*Son et al., 2012*), Nature Methods), cells originate from ATCC (Cat#CCL-219). |
| Cell line (*M. musculus*) | L1210 - ECACC | ECACC | Cat#87092804 | |
| Cell line (*M. musculus*) | Fl5.12 | Other | | Kindly provided by laboratory of Prof. Matthew Vander Heiden from MIT |
| Cell line (*M. musculus*) | BaF3 | RIKEN BioResource Center | Cat#RCB4476 | |
| Cell line (*G. gallus*) | DT40-CDK1as | Other | | Kindly provided by laboratory of Prof. Bill Earnshaw from University of Edinburgh |
| Cell line (*H. sapiens*) | S-HeLa | Other | | Kindly provided by laboratory of Kevin Elias from Brigham And Women's Hospital |
| Transfected construct (*M. musculus*) | Scrambled shRNA | VectorBuilder | Cat#LVS (VB151023-10034) | Refers to a lentiviral construct used to transfect and express the indicated shRNA. |
| Transfected construct (*M. musculus*) | 4E-BP1 shRNA #1 | VectorBuilder | Cat#LVS (VB181217-1124dqm)-C | Refers to a lentiviral construct used to transfect and express the indicated shRNA. |
| Transfected construct (*M. musculus*) | 4E-BP1 shRNA #2 | VectorBuilder | Cat#LVS (VB181217-1125ypy)-C | Refers to a lentiviral construct used to transfect and express the indicated shRNA. |
| Biological sample (*H. sapiens*) | Unpurified buffy coat for isolation of T-cells | Research Blood Components | NA | |

*Continued on next page*

*Continued*

| Reagent type (species) or resource | Designation | Source or reference | Identifiers | Additional information |
|---|---|---|---|---|
| Antibody | Phospho-Histone H3 (Ser10) (D2C8) XP Rabbit monoclonal Ab (Alexa Fluor 647 Conjugate) | Cell Signaling Technology | Cat#3458; RRID:AB_10694086 | Dilution 1/100 in PBS supplemented with 5% BSA |
| Antibody | Phospho-Histone H3 (Ser10) (D2C8) XP Rabbit monoclonal Ab (Alexa Fluor 488 Conjugate) | Cell Signaling Technology | Cat#3465; RRID:AB_10695860 | Dilution 1/100 in PBS supplemented with 5% BSA |
| Antibody | Cyclin B1 (V152) Mouse monoclonal Ab (Alexa Fluor 488 Conjugate) | Cell Signaling Technology | Cat#4112; RRID:AB_491024 | Dilution 1/50 in PBS supplemented with 5% BSA |
| Antibody | Phospho-4E-BP1 (Thr37/46) (236B4) Rabbit monoclonal Ab (PE Conjugate) | Cell Signaling Technology | Cat#7547; RRID:AB_10949897 | Dilution 1/100 in PBS supplemented with 5% BSA |
| Antibody | Phospho-S6 Ribosomal Protein (Ser235/236) (D57.2.2E) XP Rabbit monoclonal Ab (Alexa Fluor 647 Conjugate) | Cell Signaling Technology | Cat#4851; RRID:AB_10695457 | Dilution 1/100 in PBS supplemented with 5% BSA |
| Antibody | Rabbit (DA1E) monoclonal Ab IgG XP Isotype Control (PE Conjugate) | Cell Signaling Technology | Cat#5742; RRID:AB_10694219 | Dilution 1/100 in PBS supplemented with 5% BSA |
| Antibody | α-Tubulin (11H10) Rabbit monoclonal Ab (Alexa Fluor 488 Conjugate) | Cell Signaling Technology | Cat#5063; RRID:AB_10694858 | Dilution 1/200 in PBS supplemented with 5% BSA |
| Antibody | Cyclin A Mouse monoclonal Ab (H-3) (FITC Conjugate) | Santa Cruz Biotechnology | Cat#sc-271645; RRID:AB_10707658 | Dilution 1/25 in PBS supplemented with 5% BSA |
| Antibody | 4E-BP1 (53H11) Rabbit monoclonal Ab (PE Conjugate) | Cell Signaling Technology | Cat#34470 | Dilution 1/100 in PBS supplemented with 5% BSA |
| Antibody | Phospho-4EBP1 (Thr37, Thr46) Rabbit monoclonal Ab (4EB1T37T46-A5) | Thermo Fisher Scientific | Cat#MA5-27999; RRID:AB_2745012 | Dilution 1/100 in PBS supplemented with 5% BSA |
| Antibody | Goat polyclonal anti-Rabbit IgG (H + L) Cross-Adsorbed Secondary Antibody (Alexa Fluor 568 Conjugate) | Thermo Fisher Scientific | Cat#A-11011; RRID:AB_143157 | Dilution 1/1000 in PBS supplemented with 5% BSA |
| Commercial assay or kit | Click-iT Plus OPP Alexa Fluor 594 Protein Synthesis Assay Kit | Thermo Fisher Scientific | Cat#C10457 | |
| Commercial assay or kit | T-cell isolation kit | Miltenyi Biotec | Cat#130-096-495 | |
| Commercial assay or kit | Lambda Protein Phosphatase | New England Biolabs | Cat#P0753S | |
| Chemical compound, drug | O-propargyl-puromycin (OPP) | Jena Bioscience | Cat#NU-931–5; CAS:1416561-90-4 | |
| Chemical compound, drug | S-trityl-l-cysteine (STLC) | Sigma-Aldrich | Cat#164739; CAS:2799-07-7 | |
| Chemical compound, drug | Nocodazole | Sigma-Aldrich | Cat#M1404; CAS:31430-18-9 | |
| Chemical compound, drug | MG132 | Sigma-Aldrich | Cat#474787; CAS:133407-82-6 | |

*Continued on next page*

*Continued*

| Reagent type (species) or resource | Designation | Source or reference | Identifiers | Additional information |
|---|---|---|---|---|
| Chemical compound, drug | proTAME | Thermo Fisher Scientific | Cat#I44001M; CAS:1362911-19-0 | |
| Chemical compound, drug | Vinblastine | Cayman Chemical | Cat#11762; CAS:143-67-9 | |
| Chemical compound, drug | Vincristine | Cayman Chemical | Cat#11764; CAS:2068-78-2 | |
| Chemical compound, drug | TORIN-1 | Tocris Bioscience | Cat#4247; CAS: 1222998-36-8 | |
| Chemical compound, drug | RO-3306 | Cayman Chemical | Cat#15149; CAS:872573-93-8 | We observed the chemical loses its activity within a month when stored in −20°C. All experiments were done using a stock under 2 weeks old. |
| Chemical compound, drug | OTSSP167 (OTS) | Cayman Chemical | Cat#16873; CAS:1431698-10-0 | |
| Chemical compound, drug | 1NM-PP1 | Sigma-Aldrich | Cat#529581; CAS:221244-14-0 | |
| Chemical compound, drug | 4EGI-1 | Cayman Chemical | Cat#15362; CAS:315706-13-9 | |
| Chemical compound, drug | Cycloheximide (CHX) | Cayman Chemical | Cat#14126; CAS:66-81-9 | |
| Chemical compound, drug | Defactinib | Cayman Chemical | Cat#17737; CAS:1073154-85-4 | |
| Chemical compound, drug | PF-3758309 | Cayman Chemical | Cat#19186; CAS:898044-15-0 | |
| Chemical compound, drug | Nintedanib | Cayman Chemical | Cat#11022; CAS:656247-17-5 | |
| Chemical compound, drug | SNS-032 | Cayman Chemical | Cat#17904; CAS:345627-80-7 | |
| Chemical compound, drug | Tozasertib | Cayman Chemical | Cat#13600; CAS:639089-54-6 | Alternative Names : MK 0457, VX 680 |
| Chemical compound, drug | GSK 626616 | R and D Systems | Cat#6638; CAS:1025821-33-3 | |
| Chemical compound, drug | EIPA | Sigma-Aldrich | Cat#A3085; CAS:1154-25-2 | |
| Chemical compound, drug | (-)-Blebbistatin | Sigma-Aldrich | Cat#B0560; CAS:856925-71-8 | |
| Software, algorithm | MATLAB R2014b | MathWorks | | Used to analyze the SMR raw data and generate data plots. |
| Software, algorithm | OriginPro 2019 | OriginLab | | Used to perform statistical analyses and generate data plots. |
| Software, algorithm | Mass accumulation rate analysis code | This paper | | Used to analyze the SMR data. Code can be found attached to this manuscript. |

## Cell lines, primary cells and culture conditions

L1210 cells were obtained directly from ATCC, with the exception of L1210 cells shown in *Figure 2— figure supplement 2b*, which were obtained from ECACC. L1210 cells expressing the FUCCI cell cycle sensor were generated in a previous study (*Son et al., 2012*) using ATCC originating cells. The BaF3 cells were originally obtained from RIKEN BioResource Center and engineered to express BCR-ABL in a previous study (*Stevens et al., 2016*). S-HeLa cell line was a gracious gift from Dr Elias.

FL5.12 cell line was a gracious gift from Dr Vander Heiden. All DT40 cell experiments were carried out using DT40 CDK1as cell line, which was a gracious gift from Dr Samejima and Dr Earnshaw. The DT40 CDK1as cells have had their CDK1 replaced with *Xenopus laevis* CDK1 with a F80G mutation, as detailed in *Gibcus et al. (2018)*, to sensitize the CDK1 to inhibition by 1NM-PP1. Note that the CDK1as cell line is protected by MTA and the rights to this cell line belong to Prof Earnshaw and the University of Edinburgh. All cell lines tested negative for mycoplasma. Cell line identity was validated by vendors and the identity of L1210 cells from ATCC was further validated using RNA-seq data.

The L1210 cells basic experimental and culture conditions were in 2 mM L-glutamine and 11 mM glucose containing RPMI (Thermo Fisher Scientific, #11835030) supplemented with 10% FBS (Sigma-Aldrich), 1 mM sodium pyruvate (Thermo Fisher Scientific), 20 mM HEPES (Thermo Fisher Scientific) and antibiotic/antimycotic (Thermo Fisher Scientific). In *Figure 2—figure supplement 2a*, where L1210 cells were grown under different nutrient conditions, the media was kept otherwise identical except for 4% FBS and indicated concentrations of glucose (Sigma-Aldrich) or galactose (Sigma-Aldrich). The L1210 experimental conditions also included 10 nM TMRE in the media, which did not affect cell growth.

BaF3, FL5.12 and DT40 CDK1as cells were grown in media identical to L1210 cells, with the exceptions that FL5.12 cell media was supplemented with 10 ng/ml IL-3 (R and D Systems) and DT40 cell media was supplemented with 3% chicken serum (Sigma-Aldrich). S-HeLa cells were grown in DMEM (Thermo Fisher Scientific) containing 10% FBS (Sigma-Aldrich), 1 mM sodium pyruvate (Thermo Fisher Scientific), 20 mM HEPES (Thermo Fisher Scientific) and antibiotic/antimycotic (Thermo Fisher Scientific).

Human peripheral blood mononuclear cells (PBMCs), including T-cells, were isolated from unpurified buffy coat (Research Blood Components) and activated as described previously (*Cermak et al., 2016*). Briefly, PBMCs isolation was carried out using Ficoll-Paque Plus density gradient (GE) according to the manufacturer's recommendations. After isolation of the PBMC layer, the cells were subjected to red blood cell lysis using ACK lysis buffer (Thermo Fisher Scientific). The PBMCs were then washed three times and either frozen or used for isolation of specific T-cell subsets. The T-cell data shown in this paper is derived from both frozen and non-frozen samples obtained from two independent blood samples. Primary T-cells were isolated using Naive CD8 +or CD3+T Cell Isolation Kits (Miltenyi Biotec) according to supplier's instructions. The primary T-cells were activated on an anti-human CD3 (BioLegend) coated cell culture plate in identical to L1210 culture media, with the exception that the media was supplemented with 10 µM 2-mercaptoethanol, 100 U/ml IL-2 (R and D Systems) and 2 µg/ml anti-human CD28 (BioLegend). The same media was used for SMR experiments. After activation, cells were left undisturbed for 24 hr before SMR experiments, and the activation was validated by monitoring cell number and volume changes using Coulter Counter (Beckman Coulter).

The shRNA expressing L1210 cells were generated as in *Kang et al. (2019)*. L1210 cells (obtained from ATCC) were transfected using mammalian shRNA knockdown lentiviral vectors obtained from VectorBuilder Inc Each construct contained an shRNA sequence under a U6 promoter, as well as EGFP and Puro linked by T2A for selection. Full construct details can be found online at VectorBuilder.com (Vector IDs: VB190401-1106ebw, VB190401-1107bty and VB190401-1108sfm)

Control shRNA target sequence: CCTAAGGTTAAGTCGCCCTCG
4E-BP1 shRNA #1 target sequence: ATTATCTATGACCGGAAATTT (location: 233–253, CDS)
4E-BP1 shRNA #2 target sequence: CCAGTGTTTATGGTGTGATTT (location: 912–932, 3'UTR)

Transfection was carried out using 4 rounds of spinoculation. In each round, $1.5 \times 10^5$ L1210 cells were mixed with 10 µg/ml Polybrene (EMD Millipore) and approximately $1 \times 10^6$ transducing units of lentivirus. The mixture was centrifuged at 800 g for 60 min at 25°C, and the cells were moved to normal culture media. This procedure was repeated every 12 hr for a total of 4 times. 24 hr after the last round of transfection selection was started using 10 µg/ml puromycin (Sigma-Aldrich). After 5 days of selection the shRNA and EGFP expressing subpopulation was sorted out using BD FACS Aria. shRNA knockdown efficiency was validated by immunostaining 4E-BP1 and quantifying the staining levels using FACS (antibody staining details are below, *Figure 6—figure supplement 4a,b*).

## SMR device setup and experimental details

The SMR chips were fabricated as previously described (*Cermak et al., 2016*; *Son et al., 2015a*; *Son et al., 2012*) by CEA-LETI, Grenoble, France. The device setup and experimental details were

similar to those described previously (*Cermak et al., 2016*; *Son et al., 2015a*; *Son et al., 2012*). Briefly, a piezo-ceramic placed under the device vibrates the cantilever in its second flexural bending mode resonant frequency, which is typically ~1.1 MHz. The resonant frequency was monitored using piezo-resisters embedded at the base of the cantilever. A digital control platform was used to actuate the cantilever in a direct feedback mode, where an actuating signal is generated by amplifying and delaying the detected motion signal from the cantilever. Utilizing the feedback mode with a data rate of ~3000 Hz, our SMR measurement bandwidth was fixed to ~1500 Hz, which was adequate to capture fast modulating frequency signal resulting from a cell transit through the cantilever.

SMR device was operated at a fixed temperature of 37°C, by mounting the SMR chip on a copper clamp that was connected to a circulating, temperature-controlled water bath (Julabo). Fluid flow was controlled by pressure difference across the SMR input ports. Each input port was connected to a reservoir of normal cell culture media that was pressurized with 5% $CO_2$, 21% $O_2$ (Airgas) to maintain stable pH. The amount of pressure applied to each vial were controlled using electronic pressure regulators (Proportion Air QPV1) and the applied pressure difference across the channel was set to achieve a typical flow rate of ~2 nL/s to minimize the shear stress of cells growing within the SMR. This resulted in a typical ~300 ms transit time through the cantilever. Both the absolute pressure and flow direction were controlled using a custom software (LabVIEW 2012 and LabVIEW 2016). The software controls the pressure levels, and consequently the flow direction and speed, in real-time, and is automated to quickly respond to a change in resonant frequency signal. For example, a set of pressures are applied to flow a cell through the cantilever (*Figure 1—figure supplement 1a*, left #1). The resonant frequency change caused by cell transit through the cantilever automatically stops the flow (*Figure 1—figure supplement 1a*, left #2). Flow is maintained at zero for desired amount of time (~50 s), after which the pressures are changed to reverse the flow direction (*Figure 1—figure supplement 1a*, left #3). See *Figure 1—figure supplement 1* for the detailed steps of the fluid control and consequent cell movement.

For monitoring morphology changes during mitosis and measuring FUCCI cell cycle reporter intensity, we utilized an on-chip imaging system described previously (*Son et al., 2012*). Briefly, a modular Nikon microscope equipped with a Nikon LU plan ELWD 50x/0.55 objective, a Lumencor Spectra X light engine and an 8 mm Voltage Output Type photomultiplier tube (Edmund Optics) or a monochrome camera (BFS-U3-13Y3M-C, FLIR). As a cell passed through the SMR cantilever, the change in the resonance frequency was used as a trigger to turn on illumination and measure the FUCCI reporter fluorescence. On-chip cell imaging was done as in *Calistri et al. (2018)*.

## Data acquisition and processing

The motion of the cantilever and thus the resonant frequency of the SMR was measured by a digital control platform described previously (*Cermak et al., 2016*). The measured signal in digital platform was fed into custom LabVIEW code that records the signal while cell is in transit. Recorded frequency was then post-processed by custom MATLAB code, as described previously (*Cermak et al., 2016*). Briefly, the code locates two local minima in frequency peaks, fits a fourth order polynomial to the raw data, and the minimum resonance frequency values are extracted from the fittings. The average of these two resonance frequency minima measured in Hz was then transformed in to picograms by calibrating the measurements using monodisperse polystyrene beads (Thermo Fisher Scientific, Duke Standards) with a known buoyant mass. No frequency peak exclusion was performed, except in an extremely rare event where two daughter cells separate inside the cantilever during transit.

## Assigning cell cycle transitions to buoyant mass traces

We identified three distinct cell cycle transition points: mitotic entry (i.e. G2/M transition), metaphase-to-anaphase transition (i.e. M/A transition) and daughter cell abscission (*Figure 1—figure supplement 2*). We defined M-phase as the sum of mitosis and cytokinesis, starting at mitotic entry and ending at daughter cell abscission. We identified metaphase-to-anaphase transition using the mAG-Geminin signal of the FUCCI cell cycle reporter, which is degraded at metaphase-to-anaphase transition (*Figure 1—figure supplement 2c*), and using on-chip brightfield imaging, where we identified the metaphase-to-anaphase transition as the last timepoint of cell being round. These signals always coincided with two biophysical signals measured by SMR, a drop in node deviation signal (an

acoustic signal corresponding to cell shape and stiffness) and a momentary reduction in buoyant mass trace (*Figure 1—figure supplement 2b*), both partly due to cell elongation in anaphase (*Kang et al., 2019*). These elongation dependent signals are applicable to all cell types studied here and this was validated by on-chip imaging, allowing us to designate metaphase-to-anaphase transition for all cells.

The daughter abscission was assigned for all cells based on the approximately 50% loss of buoyant mass within 2 min. Cytokinesis was defined as the time between metaphase-to-anaphase transition and daughter cell abscission. G1 was defined to start immediately following cell abscission.

Detection of mitotic entry (G2/M transition) for L1210 cells was carried out using biophysical parameters. First, we have previously shown that node deviation starts to decrease following mitotic entry (*Kang et al., 2019*) (*Figure 1—figure supplement 2b*), allowing us to locate mitotic entry for L1210 traces. The timing of the assigned mitotic entry (30 min prior to metaphase-to-anaphase transition) also matched with previously analyzed mitotic entry point based on single-cell volume measurements (*Son et al., 2015a*; *Zlotek-Zlotkiewicz et al., 2015*). Similar node deviation based assignment of approximate mitotic entry was also done for other cell types, whenever node deviation changes were observable. We also estimated the approximate mitotic entry by comparing the whole cell cycle duration to that of L1210 cells. When cell cycle durations were similar, we utilized the same timing of mitotic entry (30 min prior to metaphase-to-anaphase transition) for the other cell types, which was also consistent with the timing of mitotic entry observed by node deviation measurements. While this is an approximation, it is consistent with the notion that the duration of mitosis does not vary drastically cell-to-cell (*Araujo et al., 2016*). For chemical perturbations that prolonged early mitosis, the approximate mitotic entry is separately indicated in the figures (for example, arrows in *Figure 6d*).

Approximate duration of L1210 cell metaphase was assigned using the G2/M and metaphase-to-anaphase transitions together with previous characterization of the relative durations of different mitotic stages in L1210 cells (*Son et al., 2015a*). The duration of cell elongation (i.e. anaphase) was quantified for L1210 cells previously (12 min) (*Kang et al., 2019*) and the duration was approximated for other cell types using on-chip imaging.

## Analyzing mass accumulation rate (MAR) and MAR/mass

To quantify average MAR/mass of for late G2, early mitosis, cytokinesis and newborn G1, as seen for example in *Figure 7a*, a single linear fit was made to the buoyant mass traces for each indicated cell cycle stage and the slope of the fit represents the MAR (*Figure 2—figure supplement 1a*). To minimize the error in the length of fitted segment, data points were linearly interpolated from the buoyant mass trace to accurately pinpoint the beginning and the end of the fitted segments. Then, each slope of the linear fits (MAR) was divided by the average mass during the fitting period to obtain MAR/mass.

To quantify MAR/mass dynamics within mitotic stages, as seen for example in *Figure 2a*, we quantified the slope and average mass during each 10 min segments in the buoyant mass traces (*Figure 2—figure supplement 1b*). The 10 min segments were separated by 5 min. The 10 min segments were separately fitted a linear line, and each slope of the fitted line represents MAR (*Figure 2—figure supplement 1c*). Then MAR of each 10 min segment was divided by the average mass within that 10 min segment to calculate MAR/mass (*Figure 2—figure supplement 1d*). To minimize the error in the length of fitted segment, data points were linearly interpolated from the buoyant mass trace to increase number of data points. The MAR/mass dynamics shown in *Figure 5*; *Figure 5—figure supplement 1*; *Figure 5—figure supplement 2* were processed similarly but the length of each segment and separation between each segment were reduced to 4 min and 2 min, respectively. When showing MAR/mass dynamics of individual cells (*Figure 2d*; *Figure 2—figure supplement 1c,d*; *Figure 5—figure supplement 2a*), buoyant mass traces were filtered using a moving average covering a 10 min window of data.

To correct for the cell elongation induced bias in buoyant mass measurement during cytokinesis, we performed a data correction as shown before (*Kang et al., 2019*). Briefly, the change in cell mass distribution (cell elongation) reduces the resonance frequency shift of the SMR, in a manner that is dependent on cell geometry. Using on-chip imaging, cell mass and volume information, we estimated the cell geometry to be i) spherical before elongation, ii) overlapping spheres during anaphase, and iii) spherical doublets after elongation is over (*Figure 1—figure supplement 1d,e*). With

this information we can calculate the estimated extent of the measurement bias. The details can be found in the (*Kang et al., 2019*) and in the MATLAB code attached to the supplements. The calculated extent of the measurement bias during L1210 cell cytokinesis can be seen in *Figure 1—figure supplement 1d* and in *Figure 2—figure supplement 1d*.

## Chemical perturbations

All SMR experiments with chemical perturbations of cell growth or cell cycle, apart from serial SMR experiments, were carried out by diluting the chemicals directly in to the media within SMR. Untreated cells were then loaded to the SMR for growth monitoring, so that in a typical experiment the cell was exposed to the chemical for at least 2 hr before entering mitosis. Experiments with chemical inhibitors only lasted through one cell division, so that exposure to the chemicals was always started in interphase, approximately 1–4 hr prior to mitosis. Thus, the n indicated for chemical treatments always reflects separate experiments. Control experiments were always carried out between experiments with chemical perturbations to assure that control cell growth rates were reproducible. Note that the control L1210 cell data is accumulated over several years and the growth behavior was reproducible also between different SMR devices (*Figure 2—figure supplement 2c*).

For serial SMR experiments where cells were arrested to mitosis using STLC, cells were loaded in to the serial SMR in normal culture media immediately after STLC wash off. For serial SMR experiments where cells were treated with RO-3306, cells were loaded in to the serial SMR in 1 µM RO-3306 containing culture media immediately after release from G2 arrest. In *Figure 4c*, mitotic arrests were obtained by treating unsynchronized L1210 cells with 5 µM STLC, 1 µg/ml Nocodazole, 2 µM MG132 or 20 µM proTAME.

Importantly, we observed that RO-3306 stock stored in −20°C in DMSO was not stable over several months and we therefore carried out all experiments using RO-3306 that was prepared within 1 week of the experiment.

## Cell cycle synchronizations

L1210 cells were synchronized to G2 using a double thymidine block followed by a RO3306 mediated G2 arrest. L1210 cells in confluency of $4*10^5$ cells were first treated with 2 mM Thymidine for 15 hr, then washed with PBS and moved to normal culture conditions to release from G1/S arrest. After 6 hr, 2 mM Thymidine was added for 6 hr. Cells were again washed with PBS and moved to normal culture conditions for 3 hr, after which 5 µM RO-3306 was added. 7 hr later cells were arrested in G2 (*Figure 3a*). Cells were then washed with PBS and moved to normal culture conditions (unless otherwise stated) to allow cells to uniformly progress through mitosis. Note while most cells enter mitosis soon after release from G2 arrest, some cells fail to exit G2. The release from G2 arrest was considered as time zero in protein synthesis, serial SMR and proliferation experiments.

When cells were temporarily arrested in mitosis (*Figure 7c,d,f*), cells were first synchronized to G2, then released in to 5 µM STLC. 4 hr later the cells were washed twice with media and replaced in to normal cell culture conditions. The release from STLC mediated mitotic arrest was considered as time zero in protein synthesis, serial SMR and proliferation experiments.

## Protein synthesis rate sample preparation

Protein synthesis rates were quantified using the Click-iT Plus OPP Alexa Fluor 594 Protein Synthesis Assay Kit (Thermo Fisher Scientific) together with antibody staining specific for different mitotic stages. For each replicate approximately $3*10^6$ cells were treated with 20 µM OPP for 10 min under normal culture conditions. The OPP accumulation was stopped by mixing the cells with ice cold PBS, after which the cells were quickly washed with ice cold PBS and then fixed with formaldehyde for 10 min at room temperature. To reduce the number of cell doublets fixed together, the fixation was carried out by shaking the cells on vortex in PBS and slowly adding a corresponding volume of 8% paraformaldehyde to reach a final paraformaldehyde concentration of 4%. Fixative was washed away with PBS and the cells were permeabilized using 0.5% Triton X-1000 in PBS for 10 min at room temperature. The permeabilization solution was washed away with PBS and cells were incubated in 5% BSA in PBS for 30 min to block non-specific antibody binding.

Mitotic cells were separated from interphase cells using p-Histone H3 monoclonal antibody (S10, D2C8, conjugated to Alexa 647, Cell Signaling Technology, #3458S; or S10, D2C8, conjugated to

Alexa 488, Cell Signaling Technology, #3465S). For separation of mitosis in to early and late mitosis, Cyclin B1 (V152, conjugated to Alexa 488, Cell Signaling Technology, #4112S) and Cyclin A (H-3, conjugated to FITC, Santa Cruz Biotechnology, #sc-271645 FITC) monoclonal antibodies were used. All antibody labeling steps were carried out o/n in 4°C in 5% BSA containing PBS. All antibodies were used at the concentration recommended by supplier. Antibodies were washed away using 5% BSA in PBS.

The OPP Click-IT reaction was carried out according to manufacturer's (Thermo Fisher Scientific) instructions. After OPP conjugation with Alexa Fluor 594 fluorophore, the cells were washed twice with PBS and DNA was stained for 30 min in RT with 1:2000 dilution of NuclearMask Blue (#H10325, Thermo Fisher Scientific). Finally, the cells were washed three times with PBS, mixed in to PBS supplemented with 1% BSA and put on ice until FACS analysis.

## Protein content and phosphorylation level sample preparation

To analyze the levels of total 4E-BP1 or phosphorylated S6RP and 4E-BP1, unperturbed cells were prepared using same fixation, permeabilization and blocking protocol as for protein synthesis assays. The cells were then incubated o/n in 4°C with 4E-BP1 monoclonal antibody (53H11, Cell Signaling Technology, #34470), p-4E-BP1 monoclonal antibody (Thr37/46, 236B4, conjugated to PE, Cell Signaling Technology, #7547S), p-S6RP monoclonal antibody (S235/236, D57.2.2E, conjugated to Alexa 647, Cell Signaling Technology, #4851S) or isotype specific controls (Rabbit mAb IgG, conjugated to PE or Alexa 647, Cell Signaling Technology, #5742S) in PBS solution containing 5% BSA. All antibodies were used at the concentration recommended by supplier. For analysis of total 4E-BP1 levels, cells were washed and treated with 2 µg/ml secondary antibody (Goat anti-Rabbit IgG secondary antibody conjugated to Alexa Fluor 568, #A-11011, Thermo Fisher Scientific) for 2 hr in RT. Antibodies were washed away using 5% BSA in PBS and the cells were stained for p-Histone H3 (S10) and DNA, as in protein synthesis assays. Finally, the cells were washed three times with PBS, mixed in to PBS supplemented with 1% BSA and put on ice until FACS analysis.

## Flow cytometry

FACS-based quantifications were done using BD Biosciences flow cytometer LSR II HTS with excitation lasers at 355 nm, 488 nm, 561 nm and 640 nm, and emission filters at 450/50, 530/30, 585/15 and 660/20. See (*Figure 3a,c*; *Figure 3—figure supplement 1a*) for DNA and antibody labeling that were used to separate cell cycle stages. At least 20,000 cells were analyzed for each replicate so that each analyzed subpopulation contained at least 250 cells, and typically over 1000 cells.

## Microscopy

For validation of OPP staining, L1210 cells were plated on coverslips coated with 0.1% poly-L-lysine, and prepared using same fixation, permeabilization, blocking, antibody labeling and DNA staining protocol as for protein synthesis assays. After all staining procedures were done, the cells were mounted on to microscopy slides in Vectashield mounting medium (Vector Laboratories).

For examination of phosphorylated 4E-BP1, L1210 cells were plated on coverslips coated with 0.1% poly-L-lysine, and prepared using same fixation, permeabilization and blocking protocol as for protein synthesis assays. The cells were then labeled o/n in 4°C with p-4E-BP1 monoclonal antibody (Thr37/46, 4EB1T37T46-A5, #MA5-27999, Thermo Fisher Scientific). The following day the cells were washed three times with PBS and then treated with 2 µg/ml secondary antibody (Goat anti-Rabbit IgG secondary antibody conjugated to Alexa Fluor 568, #A-11011, Thermo Fisher Scientific) for 2 hr, before an o/n labelling in 4°C with α-Tubulin monoclonal antibody (11H10, conjugated to Alexa Fluor 488, #5063, Cell Signaling Technology). The following day the cells were washed three times with PBS and DNA was stained for 30 min in RT with 1:2000 dilution of NuclearMask Blue (#H10325, Thermo Fisher Scientific). The cells were washed three times with PBS and mounted on to microscopy slides in Vectashield mounting medium (Vector Laboratories). OPP staining and 4E-BP1 phosphorylation levels were imaged using DeltaVision wide-field deconvolution microscope using standard filters (DAPI, FITC, TRITC) and 100 × objective. After deconvolution using SoftWoRx 7.0.0 software, approximately 3 µm thick section from the middle of z-slices was merged into a maximum intensity image for visualization.

Cell proliferation rate was analyzed by imaging the cells every 1 hr using IncuCyte live cell analysis imaging system by Sartorius. The relative cell count was then assessed from the phase images by analyzing the relative confluency in each sample. These values were normalized to the value at start and hourly average values were plotted together with the standard error of mean (SEM). Representative images displaying the duration of daughter cell separation (*Figure 2c,e*) were obtained in a parallel experiment using IncuCyte live cell analysis imaging system by Sartorius with 20X objective or using on-chip imaging in the SMR with the imaging setup detailed in the section '*SMR device setup and experimental details*'.

## Lambda protein phosphatase treatment

To verify the phospho-specificity of the p-4E-BP1 antibodies, cells were prepared using same fixation, permeabilization and blocking protocol as detailed above. The cells were first treated with p-Histone H3 antibody to label mitotic cells and to protect p-Histone H3 sites from Lambda phosphatase, after which the phosphatase treatment and DNA staining were carried out. The cells were then treated with 10,000 units/ml of Lambda protein phosphatase in 1X NEBuffer for protein MetalloPhosphatases that contained 1 mM $MnCl_2$ for 12 hr in 30°C, after which cells were washed twice with PBS. The Lambda phosphatase and the treatment buffer were obtained from New England BioLabs (#P0753S). Finally, DNA was stained as detailed above and the cells were analyzed using flow cytometer.

## Combining and normalizing data

Total mass accumulation for each cell was calculated from the mass accumulation traces, by assuming that the birth size of the cell was exactly half of the abscission size. Mass accumulation during mitosis (between G2/M transition and metaphase-to-anaphase transition) and during cytokinesis (between metaphase-to-anaphase transition and daughter cell abscission) was then normalized to the calculated total mass accumulation for each cell. MAR/mass traces (i.e. MAR/mass vs time) from each single cell were aligned to metaphase-to-anaphase transition. We then linearly interpolated data points from each MAR/mass trace, consequently making the total number of timepoints for each cell to be 100. Mean and SD were calculated for each timepoint and plotted as a function of approximate mitotic entry (30 min before metaphase-to-anaphase transition for most cell types). In *Figure 4*, where the drug-induced mitotic arrests inhibit us from aligning the data to metaphase-to-anaphase transition, all data was aligned to approximate mitotic entry using node deviation and MAR/mass signals. In addition, in *Figure 4d*, the MAR/mass traces were smoothed with a moving average filter of length 3. Data smoothing was not done for any other datasets. In *Figure 7a,b*, average MAR/mass for each indicated cell cycle stage (late G2, early mitosis, cytokinesis and early G1) was normalized to control values within that cell cycle stage.

## Statistical information

All statistical tests carried out, as well as descriptions of error bars and numbers of replicates, are detailed in the figure legends. All t-tests were two tailed. No replicates were excluded, except for image analysis, when cells could not be analyzed, as detailed above. No power analysis was used. Sample size was kept at or above three independent replicates, with exact sample size depending on the experimental setup. Many of the experimental approaches required time-sensitive sample processing, which limited the maximum sample size. In all FACS-based assays, the replicate number refers to independent cell cultures. In all SMR-based assays, the replicate number refers to independent cells measured through mitosis. In SMR-based assays, control samples were grown for multiple generations yielding several replicates in one experiment, whereas drug treated samples were only grown through one division so that each drug treated replicate represents a separate experiment. Experiments were repeated at least three times on separate days, unless otherwise stated in the figure legends.

The increase in L1210 cell MAR/mass from G2 to early mitosis was quantified by comparing the highest MAR/mass values observed in G2 and in early mitosis for each cell separately. The statistical comparison between these two groups was carried out by two-tailed Student's t-test.

Statistical tests were carried out using OriginPro 2019 software. The analysis of buoyant mass traces and the analysis of images was carried out using MATLAB R2014b. Visualization of microscopy images was carried out using ImageJ. All figures were compiled using Adobe Illustrator CC 2018.

## Data and data analysis code availability

All L1210 control buoyant mass measurement shown in *Figure 1* and used for quantification of MAR/mass in *Figure 2a* can be found in *Figure 1—source data 1*. This data has not been corrected for the cell elongation. Data analysis code for correcting the cell elongation bias and obtaining MAR from the buoyant mass traces can be found attached to this manuscript.

## Acknowledgements

We thank from Dr Samejima and Dr Earnshaw for providing the DT40 CDK1as cells and Dr Elias for providing the S-Hela cells. We thank E Vasile for technical support with microscopy, L Atta for assistance in isolating primary T-cells, and M Björklund, P Winter and L Mu for useful comments on the manuscript. This work was supported by Koch Institute Frontier Research Program, Koch Institute Support (core) Grant P30-CA14051 and Cancer Systems Biology Consortium U54 CA217377 from the National Cancer Institute. TPM is supported by the Wellcome Trust Sir Henry Postdoctoral Fellowship grant 110275/Z/15/Z. JHK acknowledges support from Samsung scholarship.

## Additional information

### Competing interests

Scott R Manalis: is a co-founder of Travera and Affinity Biosensors, which develops techniques relevant to the research presented. The other authors declare that no competing interests exist.

### Funding

| Funder | Grant reference number | Author |
| --- | --- | --- |
| Wellcome | 110275/Z/15/Z | Teemu P Miettinen |
| National Cancer Institute | CA217377 | Scott R Manalis |
| Koch Institute Frontier Research Program | P30-CA14051 | Scott R Manalis |
| Samsung Scholarship | | Joon Ho Kang |

The authors declare that the funders had no involvement in study design, data collection, interpretation or presentation.

### Author contributions

Teemu P Miettinen, Conceptualization, Data curation, Formal analysis, Supervision, Funding acquisition, Validation, Investigation, Visualization, Methodology, Writing—original draft, Writing—review and editing; Joon Ho Kang, Data curation, Software, Formal analysis, Validation, Investigation, Visualization, Methodology, Writing—original draft, Writing—review and editing; Lucy F Yang, Investigation, Writing—review and editing; Scott R Manalis, Resources, Supervision, Funding acquisition, Project administration, Writing—review and editing

### Author ORCIDs

Teemu P Miettinen (iD) https://orcid.org/0000-0002-5975-200X
Joon Ho Kang (iD) https://orcid.org/0000-0003-4165-7538
Lucy F Yang (iD) http://orcid.org/0000-0001-6950-7764

### Decision letter and Author response

Decision letter https://doi.org/10.7554/eLife.44700.027
Author response https://doi.org/10.7554/eLife.44700.028

## Additional files

### Supplementary files
• Source code 1. Mass accumulation rate analysis code.
DOI: https://doi.org/10.7554/eLife.44700.024
• Transparent reporting form
DOI: https://doi.org/10.7554/eLife.44700.025

### Data availability
All L1210 control buoyant mass measurement around M-phase, which were used for quantification of mitotic growth (Figure 1), MAR/mass dynamics (Figure 2), can be found in Figure 1-source data 1.

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
