## [Decision Letter]

Thank you for submitting your article "Animal cells grow during mitosis to maintain cytokinetic fidelity" for consideration by *eLife*. Your article has been reviewed by three peer reviewers, and the evaluation has been overseen by a Reviewing Editor and Jonathan Cooper as the Senior Editor. The following individuals involved in review of your submission have agreed to reveal their identity: Marvin Tanenbaum (Reviewer #2); Matthieu Piel (Reviewer #3).

The reviewers have discussed the reviews with one another and the Reviewing Editor has drafted this decision to help you prepare a revised submission.

Summary:

In this study, the authors make very sensitive single cell mass measurements to determine cellular growth during different phases of the cell cycle and find that cellular growth can also occur during (some stages of) mitosis. They go on to measure translation rates in different stages of mitosis and use a number of pharmacological inhibitors in an attempt to identify the signaling pathways that drive mitotic growth, and to study the functional consequences of inhibiting mitotic growth.

Essential revisions:

All the reviewers found your approach to be innovative and the potential of your assay to be exciting. They were all concerned, however, that several of your key conclusions (that cell growth is inhibited in metaphase, but not in prometaphase; that CDK1 acts through cap-dependent translation; that CDK acts through 4EBP1; that the cell growth promoting activity of CDK1 is needed for cell growth in G1 phase; and that the cell growth promoting activity of CDK1 is needed for symmetric cell division) are not directly supported by the data, either because of the possibility of off-target effects, or because the text and the figures appear to contradict each other. The majority of the reviewers' concerns can be addressed by re-writing the manuscript to make clear what can be directly concluded from the data, by removing the last sections of the paper concerning asymmetric cell division and 4EBP1 localisation, and, as a consequence, by changing the title.

Additional data will be required to support your conclusion on the effects on growth in G1 phase: the reviewers suggest using your single cell data to correlate, at the single cell level, the growth in mitosis and the growth in early G1, in non perturbed and in perturbed cells.

We are attaching the three reviewers concerns to guide you in amending your study.

Reviewer #1

1) Throughout the paper, what was being measured, what time-points were acquired and how the different sub-periods of mitosis should be made more explicit. A more consistent nomenclature of periods and timepoints should be used.

For example, the authors mention mass accumulation during anaphase but it is unclear how this period was defined, considering the timepoints identified on Figure 1C (it is surprising that growth can be measured during such a short event). The periods defined in Figure 1C do not have the same names as the periods on Figure 1D or Figure 2A. It was also unclear how the total cell cycle duration was measured (from the end of abscission? Or from cytokinesis completion?). Perhaps a clearer schematic could be presented that incorporates some elements of Supplementary Figure 1 to the main Figure.

Treatments applied to cells were also difficult to follow. Figure 4E,H: is the treatment done throughout the whole measurement? Figure 5C-E: Is this 1) a block to synchronize in G2, 2) a treatment with STLC, 3) a release, 4) growth measurement? (the text describing this in subsection “CDK1-driven mitotic protein synthesis supports daughter cell growth” could be made simpler). Again, schematics could be added to the figures to aid the reader.

2) Figure 2: In Figure 2D, the authors mention that MAR reaches 0 for all cell types. It looks like for S-HeLa and CD3+ T cells, MAR reaches negative values, which would mean that cells loose mass at M/A transition. Could the authors comment, and make sure that this is not a problem from measurement of adherent cells with the SMR? Figure 2A is used to document a 15.8% increase in MAR in prophase vs. late G2 but it would be better to show a box plot with <MAR> for the 2 windows of time.

3) Figure 4 is very interesting but requires clarification:

Interpretation of the effects of OTS treatment: It is hypothesized to act only via its off-target effect on CDK1, so why does it have a stronger effect on protein synthesis rate than direct inhibitors of CDK1?

Interpretation of the authors for the negative MAR in Figure 4E and H: Does it suggest that the ratio of protein degradation rate/protein synthesis rate becomes positive in the absence of CDK1-driven translation?

Some of the statements in the text could be clearer: For example, in the third paragraph of subsection “CDK1 drives phosphorylation of 4E-BP1, cap-dependent protein synthesis and mass accumulation in mitosis”, it might be more precise to say: 'cap-dependent protein synthesis persists in mitosis and partially involves CDK1-mediated regulation of cap-dependent translation'. There remains a delta in protein synthesis rate between the inhibition of translation via the inhibition of CDK1 activity using RO-3306 and the inhibition of translation via a direct inhibition of 4EBP1 (Figure 4F), thus suggesting that part of the growth in mitosis involves cap-dependent translation but via another regulatory pathway than through CDK1.

4) Figure 5, testing the recovery of growth after the release of CDK1 inhibition during mitosis is an important result that would benefit from a bit more work.

5 c and d: why are the experiments treated separately? It is surprising that the control from exp2 in 5d is similar to the treated condition in exp1. A third experimental replicate for 5c and 5d is recommended to make sure that the results are reproducible. In addition, the number of events compared in the control and treated conditions should be comparable.

The plot in 5f is supposed to show recovery of cell growth, but by examining confluency, proliferation, not growth is being measured. The size distribution of the different conditions at different time-points could prove this point (e.g. 1/2 cell cycle duration, 1 and 2 cell cycle durations after the treatment release). It would be even more interesting to add a few time points to the measurement on Figure 5E if feasible.

Some single-cell experiments datasets were relatively small compared to the control condition. Moreover, the number of replicates in Figure 5A and B was unclear and whether measurements were made on the same day (with the same batch of cells) in the control and the treated conditions. If not, a justification should be provided (e.g. at least showing reproducible MAR measurement distributions on control populations in different experiments).

5) Figure 6: is there only one biological replicate for Figure C-D?

6) In the Discussion, the authors suggest that the growth during mitosis is comparable to that of prophase. This could be more directly tested with the data the authors already have in Figure 1. The comparison of the different cell lines is interesting but could be improved by comparing the relative mass accumulation to the relative duration of mitosis of each of these cell types. Perhaps by plotting DM_mitosis/DM_total cell cycle vs. DT_mitosis/DT_total cell cycle, or measuring <DM_mitosis/DT_mitosis> and comparing with <(DM_total_cell_cycle – DM_mitosis)/(DT_total_cell_cycle – DT_mitosis)>, or doing the same boxplot as Figure 1D but sorting the cells by increasing DT_mitosis/DT_total cell cycle. Mitosis duration has been shown to function as an insulated period of the cell cycle, with very constant duration across cell types, regardless of the variety of duration of the rest of the cell cycle (see Aurajo et al., 2016). Perhaps the authors could use their data to see if the same homogeneous behavior holds true for growth. This would raise interesting hypotheses about whether general constraints on optimal energy resource allocation/protein synthesis exist during mitosis that lead to some reproducible growth behavior across cell types.

7) The statement at the start of the second paragraph of the Discussion that growth is required for cell division has not been proven. The authors do not test directly the requirement for growth to complete cell division. Doing so would require inhibiting growth (independently of the cell cycle proteins) and assessing the rate of mitotic progression (and the rate of asymmetric divisions). The authors could perform this experiment (for example using 4EBP1-I) to make this point more rigorously.

Reviewer #2

– Can the authors exclude that the morphological changes of the cell that occur during cell rounding in prometaphase affect their measurements, and perhaps cause the small bump in the graph describing cell mass in early prometaphase? Similar for the morphological changes in anaphase / telophase?

– Is the reduction in cell growth in metaphase due to metaphase itself, of simply a consequence of being in mitosis for 20-40 min? When the authors delay metaphase onset, for example with a small molecule inhibitor of Eg5, growth progressively slows or halts after 20 min of entering prometaphase. Therefore, it appears that growth does slow down after cells enter prometaphase, but the slow down is somewhat slow and therefore not complete until metaphase.

– If I'm correct, HeLa cells are the only non-lymphoid cells they use, and HeLa cells show a fairly strong reduction in growth during mitosis in their data (Figure 2D) (they don't distinguish between prometaphase and metaphase, but if looks like the growth is already reduced in PM). Their Abstract states: "Growth is only stopped in metaphase", but the HeLa cell data (As well as the DT40 cell data) appear to show a strong growth reduction already in prometaphase. Thus, there conclusions should only be limited to lymphoid cells, or the conclusion should better match the data, i.e. the observed slow down of growth in prometaphase.

– The authors are not always very accurate in their description of the different mitotic phases. For example, cells treated with STLC are in prometaphase, not in metaphase. In contrast, MG132 and proTAME to arrest cells in metaphase. Since the main focus of this study is the precise timing of cell growth during different stages of mitosis, such inaccuracies are fairly confusing.

– Based on the STLC and Vinblastine etc experiments, I belief it should be concluded that the growth inhibition does not occur specifically in metaphase, but rather occurs after entering mitosis, with some delay. Consistent with this, HeLa cells are slower in reaching anaphase, and show a greater reduction in growth before anaphase.

– Throughout the manuscript the authors claim that CDK1 stimulates translation. Yet, in late prometaphase and early metaphase there is plenty of CDK1 activity, but low translation rates. Similarly, in G2 there is low CDK1 activity, but high translation rates. So there is a fairly poor correlation between CDK1 activity and translation rates. This is poorly discussed throughout the manuscript.

– "However, treatment with both 4EGI-1 and RO-3306 did not significantly change mitotic protein synthesis rates when compared to treatment with 4EGI-1alone. Thus, cap-dependent protein synthesis persists in mitosis, and CDK1 regulates growth at least partly through cap-dependent translation."

I don't understand how the authors make this conclusion (and similar ones elsewhere in the manuscript) based on these experiments. Is the reasoning that 4EGI-1 inhibits cap-dependent translation, and in the absence of cap-dependent translation CDK1 inhibition does not reduce protein synthesis, thus CDK1 must be acting through cap-dependent translation? If so, I don't belief that the current data are solid enough to support this conclusion. 4EGI-1 treatment only reduces protein synthesis rates by 50%, both in G2 and M, so clearly there is still substantial cap-dependent translation under 4EGI-1 inhibitor treatment.

– Did the authors examine the time from mitotic entry to the metaphase-to-anaphase transition in control cells vs cells treated with a low dose of CDK1 inhibitor? It is certainly possible that chromosome alignment is perturbed under these conditions, so the time from NEB to the M-A transition is longer under partial CDK1 inhibition. All the data in this study shows that translation rates steadily decrease either from the moment of mitotic entry, or with a ~20 min delay (apparently depending on the cell line), including STLC treatment.

– Therefore, a delay in prometaphase induced by the CDK1 inhibitor would be sufficient to explain the observed decrease in the protein synthesis rates. This is especially true, because the authors align their traces at the M-A transition; if CDK1 inhibition slows down chromosome alignment, then cells that are, say 10 min before the M-A transition in control cells have been in mitosis shorter than cells 10 min before the M-A transition in CDK1 inhibited cells, thus potentially explaining why CDK1 inhibitor treatment apparently reduced translation.

– Subsection “CDK1-driven mitotic protein synthesis supports daughter cell growth”: "…the growth reduction in daughter cells must reflect a consequence of mitotic growth inhibition".

This is an overstatement, CDK1 inhibition does all kinds of things besides inhibition of growth. For example, it may delay mitosis, which activates p53, among other things and has consequences in G1 (Uetake and Sluder, 2010, Curr Biol), it could induce chromosome missegregation, or change the levels or localization of different proteins. Thus, attributing the effects of growth in daughter cells specifically to the role of CDK1 in stimulating mitotic translation or growth seems premature.

– Figure 6. Just because 4EBP1 localizes a bit more to the cleavage plane, does not prove that translation is preferentially occurring there. An OPP assay could be performed to support this claim.

– "Control cell divisions were mostly symmetrical (Figure 6C). However, when we analyzed cells treated with RO-3306, OTSSP167 or cycloheximide, we observed that all these treatments resulted in more asymmetric cell divisions"

CDK1 phosphorylates hundreds of proteins during mitosis, including many proteins that regulate microtubule and spindle function. Thus, the spindle is probably also not functioning completely normally after CDK1 inhibition, which is far more likely to be the cause of the asymmetric cell division. Therefore, I don't think that linking the observed affects on cell growth after CDK1 inhibition to the asymmetric divisions is warranted.

– "Our single-cell measurements show that mass accumulation behavior during mitosis is dynamic, dependent on mitotic stage"

I disagree that the data proves that mass accumaltion depends on the mitotic stage per se. The rate of mass accumulation appears to depend mainly on the time since entering mitosis, with a gradual shutdown of protein synthesis and growth occurring after mitotic entry. This conclusion is supported by the strong growth inhibition before the M-A transition in HeLa cells, as well as the strong growth inhibition in STLC treated cells, which never reach metaphase.

– "Indeed, we show that growth is only stopped during anaphase"

Not correct, in all cell lines, growth is already stopped in metaphase, in many cell lines, it is already strongly reduced in prometaphase.

– "Mechanistically, we found that CDK1 drives M-phase growth through 4E-BP1 and cap-dependent protein synthesis"

As stated above, I don't think this is shown. In G2 there is very low CDK1 activity, yet strong growth and translation. Similarly, in late prometaphase/metaphase there is also strong CDK1 activity, yet growth gets downregulated.

– The authors should be careful not to be selective in their choice of the different studies examining translation in mitosis to find support for their conclusions, for example:

"This is consistent with the observations that transcription and translation of

growth related genes are prioritized during mitosis". They do not discuss Stumpf et al. and Tanenbaum et al. here, who performed genome-wide analysis of translation in mitosis, and which are discussed in other parts of the manuscript.

– Title: I'm not sure that the title is sufficiently supported by the data. While the authors provide one experiment that translation is needed for cell division symmetry; cells treated with CHX (As discussed above, I don't think their inhibition of CDK1 experiments prove that cell growth in M is needed for division symmetry), they do not show that "cell growth" is needed for cell division symmetry. The authors themselves state the translation and cell growth are not the same: "Importantly, protein and RNA synthesis rates are only proxies of overall growth (biomass increase), which is determined by the balance between synthesis (anabolic) and degradation (catabolic) rates"

Reviewer #3

1) The main concern is the specificity of the effect of CDK1 inhibition on early mitotic growth. The authors show by multiple approaches that CDK1 activity is required for the maintenance of growth during early mitosis – this is fine. They then propose that it is because CDK1 phosphorylates 4E-BP1. Nevertheless, they do not formally prove this point, and CDK1 could act via other pathways. The authors should amend their text to make it clear that they did not prove that the early mitosis growth is due to CDK1 acting via 4E-BP1. They should also discuss the potential role of mitotic swelling, which they showed is also CDK1 dependent, and could affect early mitotic growth rate – when cells enter mitosis, they increase volume in a CDK1 dependent manner. One would thus expect that inhibiting CDK1 activity, which would prevent this swelling, could have an effect on overall cell density and affect the growth rate in this early mitotic phase.

2) One consequence of the previous concern is that the last piece of data provided is not really easy to interpret. It is in fact unlikely that the effect of CDK1 inhibition on the asymmetry of division is due to its effect on mitotic growth. This effect of CDK1 inhibition on mitotic spindle positioning, or in fact of almost any drug inducing a mitotic delay, as already been reported and rather depends on the importance of CDK activity for other processes that are needed to orient the spindle properly (see for example Beamish JBC 2009). I would remove this part, which does not bring much to the article and is unlikely to be correct – it is probably not because of a lack of early mitotic growth that the division ends up being asymmetrical upon CDK1 inhibition. In my opinion, the conclusion drawn “Together, these results show that CDK1 and the maintenance of active translation in M-phase are required for the fidelity of cytokinesis” is not supported by the data. But it is really not important for the rest of the article.

3) Similarly, I would remove the very short chapter on the localization of 4E-BP1. It is not clear how it helps the main message of the article.

4) To help the reader estimate the global effect of inhibiting early mitotic growth, the authors should provide, just like in Figure 1, the effective growth in mitosis (in% and not just the MAR), cytokinesis and early G1 comparing control and cells in which early mitotic growth has been inhibited.

[Editors' note: further revisions were requested prior to acceptance, as described below.]

Thank you for submitting your article "Mammalian cell growth dynamics in mitosis" for consideration by *eLife*. Your article has been reviewed by three peer reviewers, and the evaluation has been overseen by a Reviewing Editor and Jonathan Cooper as the Senior Editor. The following individuals involved in review of your submission have agreed to reveal their identity: Marvin Tanenbaum (Reviewer #2); Matthieu Piel (Reviewer #3).

The reviewers have discussed the reviews with one another and the Reviewing Editor has drafted this decision to help you prepare a revised submission. This should only require a little rewording and we look forward to your rapid revision.

Essential revisions:

The 4E-BP1 knockdown experiment does not really prove that translation in mitosis is needed for growth in G1. One could still envision a scenario in which CDK1 inhibition causes some sort of stress in G1 phase, either through an increased mitotic duration or for example through induction of chromosome missegregation (or DNA damage associated with chromosome missegregation). Such stress could trigger a translational downregulation in G1, which is (partially) restored by depletion of 4E-BP1. In this scenario it is not the role of CDK1 in mitotic translation / growth that is responsible for slower G1, but another function of CDK1. Also, the depletion of 4E-BP1 does not increase G1 translation through its activity in M phase, but through its activity in G1 phase in this scenario.

The authors should discuss this alternative possibility if they cannot fully exclude it.

---

## [Author Response]

Essential revisions:All the reviewers found your approach to be innovative and the potential of your assay to be exciting. They were all concerned, however, that several of your key conclusions (that cell growth is inhibited in metaphase, but not in prometaphase;

We have now amended our manuscript text (both Results and Discussion) to more clearly state that growth rates start to decrease in late prometaphase.

CDK1 acts through cap-dependent translation; that CDK acts through 4EBP1

4E-BP1 is a negative regulator of cap-dependent translation, and the phosphorylation of 4E-BP1 by mTOR or CDK1 inactivates 4E-BP1. We have now knocked down 4E-BP1 and monitored how this affects mitotic protein synthesis rates. Our new data show that in the absence of 4E-BP1, CDK1 inhibition no longer reduces mitotic protein synthesis as radically. Thus, this rescue experiment validates that at least part of the CDK1-driven mitotic growth is mediated by 4E-BP1. Please see our new Figure 6G, Figure 6—figure supplement 4, and paragraph four of subsection “CDK1 drives phosphorylation of 4E-BP1, protein synthesis and mass accumulation in mitosis”.

We have clarified in our Discussion that other CDK1 dependent or independent growth regulations may also exist, as our data does not rule out this possibility.

The cell growth promoting activity of CDK1 is needed for cell growth in G1 phase

Using the 4E-BP1 knockdown to rescue mitotic growth effects of CDK1 inhibition, as described above, we now show that daughter cell protein synthesis rates are less impaired in 4E-BP1 knockdown cells than in control knockdown cells. Thus, rescuing the mitotic growth inhibition also rescues the daughter cell growth inhibition. However, these rescues were not complete, and we have now clarified our text by pointing out that other mechanisms may also be involved. Please see our new Figure 7E, subsection “CDK1-driven mitotic protein synthesis supports daughter cell growth” and also the discussion of our manuscript.

In addition, as suggested by the reviewers, we have used our single cell data to correlate, at the single cell level, the growth in mitosis and the growth in early G1, in nonperturbed and in perturbed cells. This revealed a modest correlation between mitotic MAR and G1 cell MAR, and drug treatments which reduced mitotic MAR maintained a similar level of correlation. These new data can be found in our new Figure 7—figure supplement 1, and in subsection “CDK1-driven mitotic protein synthesis supports daughter cell growth”.

The cell growth promoting activity of CDK1 is needed for symmetric cell division) are not directly supported by the data, either because of the possibility of off-target effects, or because the text and the figures appear to contradict each other. The majority of the reviewers' concerns can be addressed by re-writing the manuscript to make clear what can be directly concluded from the data, by removing the last sections of the paper concerning asymmetric cell division and 4EBP1 localisation, and, as a consequence, by changing the title.

As suggested by the editor, we have removed the section regarding the asymmetrical cell divisions and subcellular localization of p4EBP1.

We have changed our manuscript title to “Mammalian cell growth dynamics in mitosis”.

Additional data will be required to support your conclusion on the effects on growth in G1 phase: the reviewers suggest using your single cell data to correlate, at the single cell level, the growth in mitosis and the growth in early G1, in non perturbed and in perturbed cells.

As suggested by the reviewers, we have now correlated our single-cell growth data non-perturbed and in perturbed cells to show a modest correlation between mitotic growth and daughter cell growth. When mitotic growth was perturbed, this correlation persisted, as also the daughter cells reduced their growth. These new data can be found in our new Figure 7—figure supplement 1, and in subsection “CDK1-driven mitotic protein synthesis supports daughter cell growth”.

As suggested by reviewer #1, we have repeated our experiments on inhibiting mitotic CDK1 activity and monitoring daughter cell growth rates using serial SMR. These new data are consistent with our previous data and our conclusion that CDK1-driven mitotic growth supports daughter cell growth. Please see Figure 7C, third panel (experiment #3).

As stated above, by knocking down 4E-BP1 we can now partly rescue the daughter cell growth impairment caused by CDK1 inhibition (Figure 7E), indicating that the mechanism regulating mitotic growth is also affecting daughter cell growth.

Other major changes:

We have taken advantage of the free *eLife* manuscript format by reorganizing some of our data. In the current organization, each figure has a more specific message and content, which will make our manuscript more accessible to readers.

As requested by reviewers #2 and #3, we have now addressed how morphological changes (mitotic swelling, rounding and cytokinesis) affect our data. We have now added several completely new figures (Figure 5A,B; Figure 5—figure supplement 1; Figure 5—figure supplement 2) to our manuscript. Experiments in these figures show that when we inhibit mitotic swelling or cytokinetic cell elongation the mitotic MAR dynamics are not radically altered. Thus, the morphological changes in mitosis do not explain the mitotic growth dynamics. Please see subsection “Cells in metaphase and anaphase display mitotic stage specific inhibition of mass accumulation” for full details.

As requested by reviewer #2, we have now studied if mitotic growth rate is purely dependent on the time that cells have spent in mitosis or if there are additional growth reducing mechanism(s) in later mitotic stages (metaphase and anaphase) that inhibit growth. In our new figures (Figure 5C,D; Figure 5—figure supplement 2C) we show that time after mitotic entry does not alone explain the cell growth rate decrease observed during control cell mitosis. Please see subsection “Cells in metaphase and anaphase display mitotic stage specific inhibition of mass accumulation”, as well as our revised Discussion section, for full details.

We are attaching the three reviewers' concerns to guide you in amending your study.Reviewer #11) Throughout the paper, what was being measured, what time-points were acquired and how the different sub-periods of mitosis should be made more explicit. A more consistent nomenclature of periods and timepoints should be used.

Thank you for pointing out these important shortcomings. We have now added clarity to the text in the beginning of our Results section to better define early mitosis (from mitotic entry to metaphase-to-anaphase transition) and cytokinesis (from metaphase-to-anaphase transition to the abscission of daughter cells). We also more clearly state that G2/M transition is the same as mitotic entry. In addition, our Material and methods section has been amended to clarify the points raised by the reviewer.

For example, the authors mention mass accumulation during anaphase but it is unclear how this period was defined, considering the timepoints identified on Figure 1C (it is surprising that growth can be measured during such a short event).

We have now clarified in our main text and Materials and methods section that anaphase refers to the approximate duration of cell elongation (anaphase and telophase overlap in most cells studied).

The periods defined in Figure 1C do not have the same names as the periods on Figure 1D or Figure 2A.

We have changed the naming in Figure 1C to correspond to Figures 1D and 2A. We have also clarified in the beginning of our Results section the terminology used.

It was also unclear how the total cell cycle duration was measured (from the end of abscission? Or from cytokinesis completion?).

We measure the total cell cycle duration by quantifying the duration between successive abscissions (see Figure 1B for an example where the blue arrows indicate the abscissions). We also use the doubling time in cell culture to validate the cell cycle time for the cell types for which we have limited single-cell data.

Perhaps a clearer schematic could be presented that incorporates some elements of Supplementary Figure 1 to the main Figure.

We have incorporated color bars to Figure 1C which correspond to the durations quantified in Figure 1D (i.e. early mitosis and cytokinesis).

Treatments applied to cells were also difficult to follow. Figure 4E,H: is the treatment done throughout the whole measurement?

Thank you for pointing this out. The treatments were applied to the cells 1-4 hours prior to mitotic entry, and the treatments were maintained throughout the whole measurement. We have now clarified this in the figure legends.

Figure 5C-E: Is this 1) a block to synchronize in G2, 2) a treatment with STLC, 3) a release, 4) growth measurement? (the text describing this in subsection “CDK1-driven mitotic protein synthesis supports daughter cell growth” could be made simpler). Again, schematics could be added to the figures to aid the reader.

We have now added a schematic of the workflow to the Figure 7—figure supplement 2A. We have also clarified our Results section “CDK1-driven mitotic protein synthesis supports daughter cell growth” to make this clearer.

2) Figure 2: In Figure 2D, the authors mention that MAR reaches 0 for all cell types. It looks like for S-HeLa and CD3+ T cells, MAR reaches negative values, which would mean that cells loose mass at M/A transition. Could the authors comment, and make sure that this is not a problem from measurement of adherent cells with the SMR?

The reviewer raises an important point. Most importantly, this negative MAR is not due to measurement bias, but reflects a genuine and common feature for many cells. More specifically, we show that:

The cell elongation in anaphase can induce a small bias in our mass measurements. We have now corrected this in all our data and figures using a correction method which we recently published (Kang et al., 2019). Examples comparing corrected and non-corrected data can be found in Figure 1—figure supplement 1E; Figure 2—figure supplement 1D. This correction does not remove the negative MAR observed in some cells, as our buoyant mass measurement is only affected by less than 1%.

If cytokinesis is block without inhibiting mitosis, and thus cell elongation does not take place, MAR can still become negative around the metaphase-to-anaphase transition (please see Figure 5—figure supplement 2B, where we inhibited cytokinesis using blebbistatin).

The negative MAR is more pronounced and observed in more cells if we shorten the window in which we analyze MAR (i.e. increase temporal resolution and sensitivity in MAR analysis). Please see Figure 5B, which displays that also L1210 cells can display negative MAR.

We have now pointed out to the reader in our Results section that we observe this negative MAR, and we have added a paragraph to the Discussion that explain how MAR reflects the flux of components (e.g. nutrients) in and out of the cell. Thus, this negative MAR may arise from, for example, cell blebbing and exocytosis during anaphase, but as we have evidence for this, we do not want to speculate on it.

Figure 2A is used to document a 15.8% increase in MAR in prophase vs. late G2 but it would be better to show a box plot with <MAR> for the 2 windows of time.

We have now added a violin plot to show this MAR increase. Please see our new Figure 2B.

3) Figure 4 is very interesting but requires clarification:Interpretation of the effects of OTS treatment: It is hypothesized to act only via its off-target effect on CDK1, so why does it have a stronger effect on protein synthesis rate than direct inhibitors of CDK1?

CDK1 is one potential candidate for the mechanisms behind growth reduction upon OTS treatment, as CDK1 is a known OTS off-target (Klaeger et al., 2017, Science). Importantly, this does not exclude other (additional) mechanisms. We have now removed all suggestions that CDK1 is the mediator of OTS effects from our main text and we only point out in the supplementary figure legends that CDK1 is an OTS off-target. Even though we do not know the mechanisms, we wanted to include the OTS data in our manuscript. Previously, the MELK kinase, which is the designated target of the OTS drug, has been implicated in mitotic growth (Wang et al., 2016). Yet, our experiments suggest that MELK is not the mechanisms through which OTS affects mitotic growth since other MELK kinase inhibitors did not show mitosis specific effects on protein synthesis.

Interpretation of the authors for the negative MAR in Figure 4E and H: Does it suggest that the ratio of protein degradation rate/protein synthesis rate becomes positive in the absence of CDK1-driven translation?

We do not have direct evidence on how the protein degradation/synthesis ratio changes in the absence of CDK1-driven translation. However, we have further improved our Result and Discussion section to discuss the negative MAR in more detail (as pointed out above) and to clarify that MAR does not always reflect protein synthesis rates. Please see our updated Discussion for details.

Some of the statements in the text could be clearer: For example, in the third paragraph of subsection “CDK1 drives phosphorylation of 4E-BP1, cap-dependent protein synthesis and mass accumulation in mitosis”, it might be more precise to say: 'cap-dependent protein synthesis persists in mitosis and partially involves CDK1-mediated regulation of cap-dependent translation'. There remains a delta in protein synthesis rate between the inhibition of translation via the inhibition of CDK1 activity using RO-3306 and the inhibition of translation via a direct inhibition of 4EBP1 (Figure 4F), thus suggesting that part of the growth in mitosis involves cap-dependent translation but via another regulatory pathway than through CDK1.

The delta in protein synthesis rate in our old Figure 4F (Figure 6H in our updated manuscript) may be due to the fact that we cannot completely inhibit CDK1 as complete inhibition of CDK1 would prevent cells from entering mitosis. However, we acknowledge our initial statements may have been too strong and we have now amended the text to increase clarity and avoid any overstatements.

4) Figure 5, testing the recovery of growth after the release of CDK1 inhibition during mitosis is an important result that would benefit from a bit more work.5 c and d: why are the experiments treated separately? It is surprising that the control from exp2 in 5d is similar to the treated condition in exp1. A third experimental replicate for 5c and 5d is recommended to make sure that the results are reproducible. In addition, the number of events compared in the control and treated conditions should be comparable.

The two experiments were shown separately because of the difference in control cell growth rates. As requested by the reviewer, we have now added a third experimental repeat to show that the conclusion (partial CDK1 inhibition in mitosis results in degreased growth rate in daughter cells) is reproducible (Figure 7C, third panel; experiment #3). The number of cells we measure depends on multiple factors, not all of which we have control (For example, channel clogging in the middle of experiment). We have carried out the experiments by using the same experimental time (time after release from cell cycle arrest) to ensure that control and treated cells are comparable (despite the occasional difference in cell numbers).

The plot in 5f is supposed to show recovery of cell growth, but by examining confluency, proliferation, not growth is being measured. The size distribution of the different conditions at different time-points could prove this point (e.g. 1/2 cell cycle duration, 1 and 2 cell cycle durations after the treatment release). It would be even more interesting to add a few time points to the measurement on Figure 5E if feasible.

We agree with the reviewer that confluency is dependent on proliferation. However, in our model proliferation is not fully independent from growth, since proliferation (i.e. doubling time) is correlated with overall growth. To better address the reviewer’s comment, we have carried out two experiments:

As suggested by the reviewer, we have examined G1 cell protein synthesis rates at two time points after mitosis (Figure 7E). We do observe partial recovery of protein synthesis. However, as the cell cycle synchrony is quickly lost, and not all G2 arrested cells progress through mitosis (see our Figure 3A for an example), later time points cannot reliably separate the cells that have undergone mitosis in the presence of mitotic growth inhibition. Thus, the daughter cell growth may also recover more, but our measurements cannot resolve this.

As the reviewer also suggested, we have measured the size distribution of the different conditions at different time points. If we could achieve a perfect cell synchrony, this could reveal volume distribution shifting over time. Unfortunately, when we perturb mitotic growth (whether by temporary mitotic arrest or by partial CDK1 inhibition) we also interfere with the synchrony, as these treatments reduce the% of cells that eventually divide. Consequently, for example the sample that was released from temporary mitotic arrests will display larger average cell size than control sample at 6h timepoint after release because some of the treated cells are not dividing. Thus, we can only say that all conditions return back to normal size distribution 50 hours after release from G2 arrest (see Author response image 1). Because of the limited information that this experiment provides, we do not feel these results are suitable to be presented in our manuscript.

**Author response image 1. respfig1:** Cell size histograms of control (blue), RO-3306 (red) and STLC (yellow) treated L1210 cells 50 h after release from G2 (or mitotic for STLC) arrest.

Some single-cell experiments datasets were relatively small compared to the control condition. Moreover, the number of replicates in Figure 5A and B was unclear and whether measurements were made on the same day (with the same batch of cells) in the control and the treated conditions. If not, a justification should be provided (e.g. at least showing reproducible MAR measurement distributions on control populations in different experiments).

Thank you for pointing out this lack of clarity in our manuscript. All the replicates for all the drug treatments represent completely independent experiments (i.e. we only grow the cells through one cell division). For controls, where growth is not perturbed, we typically grow the cells through multiple cell division. Due to the low throughput of our measurements, the measurements typically represent data collected over multiple months (for drug treatments) or years (for controls). We have always carried out control experiments between the drug treatment experiments to ensure that the cell growth behavior is reproducible. As requested by the reviewer, we have now added a figure displaying the reproducibility of control experiments to Figure 2—figure supplement 2C. In addition, we have clarified the measurement details and measurement reproducibility in the Methods section.

5) Figure 6: is there only one biological replicate for Figure C-D?

No, the n referred to independent experiments for drug treated cells. But, as suggested by the editors, we have now removed the Figure 6 and all data relating to the asymmetric cell division and the localization of p-4E-BP1.

6) In the Discussion, the authors suggest that the growth during mitosis is comparable to that of prophase. This could be more directly tested with the data the authors already have in Figure 1. The comparison of the different cell lines is interesting but could be improved by comparing the relative mass accumulation to the relative duration of mitosis of each of these cell types. Perhaps by plotting DM_mitosis/DM_total cell cycle vs. DT_mitosis/DT_total cell cycle, or measuring <DM_mitosis/DT_mitosis> and comparing with <(DM_total_cell_cycle – DM_mitosis)/(DT_total_cell_cycle – DT_mitosis)>, or doing the same boxplot as Figure 1D but sorting the cells by increasing DT_mitosis/DT_total cell cycle. Mitosis duration has been shown to function as an insulated period of the cell cycle, with very constant duration across cell types, regardless of the variety of duration of the rest of the cell cycle (see Aurajo et al,. 2016). Perhaps the authors could use their data to see if the same homogeneous behavior holds true for growth. This would raise interesting hypotheses about whether general constraints on optimal energy resource allocation/protein synthesis exist during mitosis that lead to some reproducible growth behavior across cell types.

Thank you for the interesting suggestion. Unfortunately, apart from S-HeLa cells, all the cell lines we analyzed have near identical duration of early mitosis (G2/M transition to M/A transition) and of the whole cell cycle. Thus, all the cell lines will have more or less the same relative duration of mitosis (this includes S-HeLa cells) and, indeed, all the cell lines display similar amount of growth in early mitosis (Figure 1D). We feel that the comparison suggested by the reviewer will only be useful if more cell lines with more variable mitotic or cell cycle durations are analyzed. Instead, we have now pointed out in our Figure 1D legend that the relative growth in early mitosis is very similar between all tested cell types.

7) The statement at the start of the second paragraph of the Discussion that growth is required for cell division has not been proven. The authors do not test directly the requirement for growth to complete cell division. Doing so would require inhibiting growth (independently of the cell cycle proteins) and assessing the rate of mitotic progression (and the rate of asymmetric divisions). The authors could perform this experiment (for example using 4EBP1-I) to make this point more rigorously.

As suggested by the editor, we have now removed the Figure 6 and all data relating to the asymmetric cell division and the localization of p-4E-BP1.

Reviewer #2

One of the main points raised by the reviewer at several occasions is that mitotic growth rate may depend purely on the time spent in mitosis. The reviewer suggests that this could explain why CDK1 inhibitions (which prolong mitosis) and mitotic arrests reduce growth rate. As stated in the major changes section at the beginning of our response letter: We have now studied if mitotic growth rate is purely dependent on the time cells have spent in mitosis or if additional growth reducing mechanism(s), that are specific to later mitotic stages (metaphase and anaphase), inhibit mitotic growth. In our new main figure (Figure 5C,D; Figure 5—figure supplement 2C) we study how growth rate depends on the time cells have spent in mitosis. By comparing the MAR of control cells and prometaphase arrested cells we show that in prometaphase both samples display similar MAR. However, as control cells proceed to metaphase and anaphase, their MAR decreases lower than what is seen in prometaphase arrested cells, although both samples have spent the same time in mitosis. Thus, time after mitotic entry alone does not explain the cell growth rate decrease observed during control cell mitosis. Therefore, additional mechanism(s) have to exist to reduce growth in metaphase and anaphase. Yet, the reviewer is correct that time after mitosis can, and probably will affect mitotic growth rate, as suggested by mitotic arrests. We have now clarified this in our manuscript writing. Please see subsection “Cells in metaphase and anaphase display mitotic stage specific inhibition of mass accumulation” of our Results section as well as our updated Discussion section for full details.

– Can the authors exclude that the morphological changes of the cell that occur during cell rounding in prometaphase affect their measurements, and perhaps cause the small bump in the graph describing cell mass in early prometaphase? Similar for the morphological changes in anaphase / telophase?

The reviewer raises an important concern here. All cell types presented in this work have spherical shapes throughout their whole cell cycle except cytokinesis, therefore making the cell shape changes in prophase minimal. However, cells do increase their volume (~10-20%) during early mitosis, where mitotic cell swelling takes place (Son et al., 2015). As stated in the major changes section at the beginning of our response letter:

We have now addressed how morphological changes (mitotic swelling, rounding and cytokinesis) affect our data. We have added a completely new figures (Figure 5; Figure 5—figure supplement 1,2) to our manuscript, in which we study mitotic growth dynamics when we inhibit mitotic swelling and cytokinetic cell elongation. Although the mitotic swelling is responsible for the increased MAR observed in early mitosis, it does not explain the rest of the MAR dynamics. Furthermore, mitotic protein synthesis dynamics did not display any change when swelling was inhibited. Inhibition of cytokinesis did not change MAR dynamics. Please see subsection “Cells in metaphase and anaphase display mitotic stage specific inhibition of mass accumulation” for full details.

In addition, we have identified that the cell elongation in anaphase can induce a small bias in our mass measurements. We have now corrected this in all our data and figures using a correction method which we recently published (Kang et al., 2019). Examples comparing corrected and non-corrected data can be found in Figure 1—figure supplement 1E; Figure 2—figure supplement 1D. This correction does not remove the negative MAR observed in some cells (our buoyant mass measurement is only affected by <1%).

– Is the reduction in cell growth in metaphase due to metaphase itself, of simply a consequence of being in mitosis for 20-40 min? When the authors delay metaphase onset, for example with a small molecule inhibitor of Eg5, growth progressively slows or halts after 20 min of entering prometaphase. Therefore, it appears that growth does slow down after cells enter prometaphase, but the slow down is somewhat slow and therefore not complete until metaphase.

The reviewer raises an important point, which we address in the beginning of this point-by-point response.

– If I'm correct, HeLa cells are the only non-lymphoid cells they use, and HeLa cells show a fairly strong reduction in growth during mitosis in their data (Figure 2D) (they don't distinguish between prometaphase and metaphase, but if looks like the growth is already reduced in PM). Their Abstract states: "Growth is only stopped in metaphase", but the HeLa cell data (As well as the DT40 cell data) appear to show a strong growth reduction already in prometaphase. Thus, there conclusions should only be limited to lymphoid cells, or the conclusion should better match the data, i.e. the observed slow down of growth in prometaphase.

Thank you for pointing this out. Our original idea with this sentence was that growth being stopped means that MAR becomes (near) zero, but we now realize this may be misleading to the readers. Since we don’t optically monitor the chromosome condensation and alignment of each cell, we cannot pinpoint the exact prometaphase-to-metaphase transition. Also, we don’t have information on the relative duration of each mitotic phase in S-HeLa or DT40 cells, so we cannot conclude if some of the growth reduction would take place already late prometaphase. We have therefore amended our Abstract by stating that growth rates decrease as cells approach metaphase-to-anaphase transition (to avoid claims which we cannot fully validate). Then, in our Results section we specify that this reduction of growth may start in late prometaphase. We have also specified in our results and Discussion sections that our results reflect the growth behavior of suspension grown animal cells.

– The authors are not always very accurate in their description of the different mitotic phases. For example, cells treated with STLC are in prometaphase, not in metaphase. In contrast, MG132 and proTAME to arrest cells in metaphase. Since the main focus of this study is the precise timing of cell growth during different stages of mitosis, such inaccuracies are fairly confusing.

Thank you for clarifying this. We have now amended our text by stating that STLC arrests the cells in a prometaphase state.

– Based on the STLC and Vinblastine etc experiments, I belief it should be concluded that the growth inhibition does not occur specifically in metaphase, but rather occurs after entering mitosis, with some delay. Consistent with this, HeLa cells are slower in reaching anaphase, and show a greater reduction in growth before anaphase.

The reviewer raises an important point, which we address in the beginning of this point-by-point response.

– Throughout the manuscript the authors claim that CDK1 stimulates translation. Yet, in late prometaphase and early metaphase there is plenty of CDK1 activity, but low translation rates. Similarly, in G2 there is low CDK1 activity, but high translation rates. So there is a fairly poor correlation between CDK1 activity and translation rates. This is poorly discussed throughout the manuscript.

Thank you for pointing this out. In G2, for example, signaling components like mTOR are well-known to promote growth through 4E-BP1 and cap-dependent translation. Our data is consistent with this, as inhibition of mTOR reduces protein synthesis and 4E-BP1 phosphorylation in G2. During mitosis CDK1 has been shown to substitute for the role of mTOR (Shuda et al., 2015), which is consistent with our data or CDK1 being required for the phosphorylation of 4E-BP1 and protein synthesis in mitosis. The reviewer is correct that around late prometaphase and metaphase growth slows down although CDK1 remains active. We have now clarified our writing that additional mechanism(s) are needed to explain this decrease in growth rates (please see our updated Discussion). Furthermore, we have provided experimental evidence that CDK1 is responsible for promoting growth in mitosis. As stated in the major changes section at the beginning of our response letter:

4E-BP1 is a negative regulator of cap-dependent translation, and the phosphorylation of 4E-BP1 by mTOR or CDK1 inactivates 4E-BP1. We have now knocked down 4E-BP1 and monitored how this affects mitotic protein synthesis rates. Our new data shows that in the absence of 4E-BP1 CDK1 inhibition no longer reduces mitotic protein synthesis as radically. Thus, this rescue experiment validates that at least part of the CDK1 driven mitotic growth is mediated by 4E-BP1. These new data can be found in our new Figure 6G, Figure 6—figure supplement 4, and in subsection “CDK1 drives phosphorylation of 4E-BP1, protein synthesis and mass accumulation in mitosis”. We have also clarified in our Discussion that other CDK1 dependent and independent growth regulations may also exist, as our data does not rule this out.

– "However, treatment with both 4EGI-1 and RO-3306 did not significantly change mitotic protein synthesis rates when compared to treatment with 4EGI-1alone. Thus, cap-dependent protein synthesis persists in mitosis, and CDK1 regulates growth at least partly through cap-dependent translation."I don't understand how the authors make this conclusion (and similar ones elsewhere in the manuscript) based on these experiments. Is the reasoning that 4EGI-1 inhibits cap-dependent translation, and in the absence of cap-dependent translation CDK1 inhibition does not reduce protein synthesis, thus CDK1 must be acting through cap-dependent translation? If so, I don't belief that the current data are solid enough to support this conclusion. 4EGI-1 treatment only reduces protein synthesis rates by 50%, both in G2 and M, so clearly there is still substantial cap-dependent translation under 4EGI-1 inhibitor treatment.

The reviewer is correct about the logic of our work (when cap-dependent translation is inhibited, then partial inhibition of CDK1 no longer influences mitotic protein synthesis, as would be expected if CDK1 promotes growth through cap-dependent translation). The fact that approximately 50% of the protein synthesis rates persist in the presence of 4EGI-1 does not make this experiment invalid. However, we do now point out to the reader that 4EGI-1 reduced translation rates approximately 50%. In addition, our new experimental evidence detailed in the major changes section at the beginning of our response letter (and in the previous answer) shows that CDK1 is at least partly promoting translation through 4E-BP1, a controller of cap-dependent translation. While we believe that this data validates our claims that “CDK1 regulates growth at least partly through cap-dependent translation”, we have also changed our manuscript text to more clearly point out that we cannot rule out other CDK1 dependent and independent mechanisms (please see our updated Discussion section).

– Did the authors examine the time from mitotic entry to the metaphase-to-anaphase transition in control cells vs cells treated with a low dose of CDK1 inhibitor? It is certainly possible that chromosome alignment is perturbed under these conditions, so the time from NEB to the M-A transition is longer under partial CDK1 inhibition. All the data in this study shows that translation rates steadily decrease either from the moment of mitotic entry, or with a ~20 min delay (apparently depending on the cell line), including STLC treatment.– Therefore, a delay in prometaphase induced by the CDK1 inhibitor would be sufficient to explain the observed decrease in the protein synthesis rates. This is especially true, because the authors align their traces at the M-A transition; if CDK1 inhibition slows down chromosome alignment, then cells that are, say 10 min before the M-A transition in control cells have been in mitosis shorter than cells 10 min before the M-A transition in CDK1 inhibited cells, thus potentially explaining why CDK1 inhibitor treatment apparently reduced translation.

Again, the reviewer raises a good point. We have examined the time from mitotic entry to metaphase-anaphase transition in cells treated with RO3306, and the reviewer is correct that the duration of early mitosis is increased under partial CDK1 inhibition, especially in the L1210 cells. However, this alone does not explain the reduced growth in early mitosis, because partial CDK1 inhibition also removes the growth rate increase seen in control cells after mitotic entry (note that mitotic arrests, like STLC treatment, still display an increased growth during early mitosis). In addition, as discussed above, we have now knocked down 4E-BP1 and show that this partly rescues RO3306 mediated mitotic growth inhibition, indicating that CDK1 regulates growth at least partly through 4E-BP1 (and not only by lengthening mitosis). Importantly, this does not rule out the option that reviewer suggests (i.e. time after mitotic entry could, and probably will affect growth rate). We have now clarified to the reader that partial CDK1 inhibition also increased the duration of early mitosis (subsection “CDK1 drives phosphorylation of 4E-BP1, protein synthesis and mass accumulation in mitosis” of our Results section) and we point out the mitotic entry in the MAR traces where CDK1 is inhibited (please see arrows in Figures 6D,F).

– Subsection “CDK1-driven mitotic protein synthesis supports daughter cell growth”: "…the growth reduction in daughter cells must reflect a consequence of mitotic growth inhibition".This is an overstatement, CDK1 inhibition does all kinds of things besides inhibition of growth. For example, it may delay mitosis, which activates p53, among other things and has consequences in G1 (Uetake and Sluder, 2010, Curr Biol), it could induce chromosome missegregation, or change the levels or localization of different proteins. Thus, attributing the effects of growth in daughter cells specifically to the role of CDK1 in stimulating mitotic translation or growth seems premature.

Thank you for pointing this out. We realize that our original statement was too bold and thus we have now softened our claims. However, we have also provided additional data to strengthen the relationship between CDK1 driven mitotic growth and daughter cell growth. As stated in the major changes section at the beginning of our response letter:

Using the 4E-BP1 knockdown to rescue mitotic growth effects of CDK1 inhibition, we now show that daughter cell protein synthesis rates are less impaired in 4E-BP1 knockdown cells than in control knockdown cells. Thus, rescuing the mitotic growth inhibition also rescues the daughter cell growth inhibition. However, these rescues were not complete, and we have now clarified our text by pointing out that other mechanisms may also be involved. Please see our new Figure 7E, subsection “CDK1-driven mitotic protein synthesis supports daughter cell growth” and also the discussion of our manuscript.

– Figure 6. Just because 4EBP1 localizes a bit more to the cleavage plane, does not prove that translation is preferentially occurring there. An OPP assay could be performed to support this claim.

As suggested by the editor, we have now removed the Figure 6 and all data relating to the asymmetric cell division and the localization of p-4E-BP1.

– "Control cell divisions were mostly symmetrical (Figure 6C). However, when we analyzed cells treated with RO-3306, OTSSP167 or cycloheximide, we observed that all these treatments resulted in more asymmetric cell divisions"CDK1 phosphorylates hundreds of proteins during mitosis, including many proteins that regulate microtubule and spindle function. Thus, the spindle is probably also not functioning completely normally after CDK1 inhibition, which is far more likely to be the cause of the asymmetric cell division. Therefore, I don't think that linking the observed affects on cell growth after CDK1 inhibition to the asymmetric divisions is warranted.

As suggested by the editor, we have now removed the Figure 6 and all data relating to the asymmetric cell division and the localization of p-4E-BP1.

– "Our single-cell measurements show that mass accumulation behavior during mitosis is dynamic, dependent on mitotic stage"I disagree that the data proves that mass accumulation depends on the mitotic stage per se. The rate of mass accumulation appears to depend mainly on the time since entering mitosis, with a gradual shutdown of protein synthesis and growth occurring after mitotic entry. This conclusion is supported by the strong growth inhibition before the M-A transition in HeLa cells, as well as the strong growth inhibition in STLC treated cells, which never reach metaphase.

The reviewer raises an important point, which we address in the beginning of this point-by-point response.

– "Indeed, we show that growth is only stopped during anaphase"Not correct, in all cell lines, growth is already stopped in metaphase, in many cell lines, it is already strongly reduced in prometaphase.

Thank you for pointing out the different interpretation of the data and text. As stated above, since we don’t optically monitor the chromosome condensation and alignment of each cell, we cannot pinpoint the exact prometaphase-to-metaphase transition. Also, we don’t have information on the relative duration of each mitotic phase in S-HeLa or DT40 cells, so we cannot conclude if some of the growth reduction would take place already late prometaphase. We have therefore amended our Abstract by stating that growth rates decrease as cells approach metaphase-to-anaphase transition (to avoid claims which we cannot fully validate). Then, in our Results section we specify that this reduction of growth may start in late prometaphase.

– "Mechanistically, we found that CDK1 drives M-phase growth through 4E-BP1 and cap-dependent protein synthesis"As stated above, I don't think this is shown. In G2 there is very low CDK1 activity, yet strong growth and translation. Similarly, in late prometaphase/metaphase there is also strong CDK1 activity, yet growth gets downregulated.

As stated above: In G2 signaling components like mTOR are well-known to promote growth through 4E-BP1 and cap-dependent translation. Our data is consistent with this, as inhibition of mTOR reduces protein synthesis and 4E-BP1 phosphorylation in G2. During mitosis CDK1 has been shown to substitute for the role of mTOR (Shuda et al., 2015), which is consistent with our data or CDK1 being required for the phosphorylation of 4E-BP1 and protein synthesis in mitosis. Now, the reviewer is correct that around late prometaphase and metaphase growth slows down although CDK1 remains active. We have now clarified our writing that additional mechanism(s) are needed to explain this decrease in growth rates. Furthermore, we have provided experimental evidence that CDK1 is responsible for promoting growth in mitosis. As stated in the major changes section at the beginning of our response letter:

4E-BP1 is a negative regulator of cap-dependent translation, and the phosphorylation of 4E-BP1 by mTOR or CDK1 inactivates 4E-BP1. We have now knocked down 4E-BP1 and monitored how this affects mitotic protein synthesis rates. Our new data shows that in the absence of 4E-BP1 CDK1 inhibition no longer reduces mitotic protein synthesis as radically. Thus, this rescue experiment validates that at least part of the CDK1 driven mitotic growth is mediated by 4E-BP1. Please see our new Figure 6G, Figure 6—figure supplement 4, and subsection “CDK1 drives phosphorylation of 4E-BP1, protein synthesis and mass accumulation in mitosis”.

We have clarified in our Discussion that other CDK1 dependent and independent growth regulations may also exist, as our data does not rule this out.

– The authors should be careful not to be selective in their choice of the different studies examining translation in mitosis to find support for their conclusions, for example:"This is consistent with the observations that transcription and translation ofgrowth related genes are prioritized during mitosis". They do not discuss Stumpf et al. and Tanenbaum et al. here, who performed genome-wide analysis of translation in mitosis, and which are discussed in other parts of the manuscript.

Thank you for pointing out this shortcoming of our writing. We have now added a note to our Discussion stating that “it should be noted that the active mitotic translation of growth promoting components remains controversial (Stumpf et al., 2013; Tanenbaum et al., 2015)”.

– Title: I'm not sure that the title is sufficiently supported by the data. While the authors provide one experiment that translation is needed for cell division symmetry; cells treated with CHX (As discussed above, I don't think their inhibition of CDK1 experiments prove that cell growth in M is needed for division symmetry), they do not show that "cell growth" is needed for cell division symmetry. The authors themselves state the translation and cell growth are not the same: "Importantly, protein and RNA synthesis rates are only proxies of overall growth (biomass increase), which is determined by the balance between synthesis (anabolic) and degradation (catabolic) rates"

We understand that concerns raised by the reviewer and, as suggested by the editors, we have now removed our last main figure and all data relating to the asymmetric cell division and the localization of p-4E-BP1. Consequently, we have changed our title to “Mammalian cell growth dynamics in mitosis”.

Reviewer #31) The main concern is the specificity of the effect of CDK1 inhibition on early mitotic growth. The authors show by multiple approaches that CDK1 activity is required for the maintenance of growth during early mitosis – this is fine. They then propose that it is because CDK1 phosphorylates 4E-BP1. Nevertheless, they do not formally prove this point, and CDK1 could act via other pathways. The authors should amend their text to make it clear that they did not prove that the early mitosis growth is due to CDK1 acting via 4E-BP1.

The reviewer raises an important concern. We have now addressed this concern both experimentally and by rewriting our manuscript, as stated in the major changes section at the beginning of our response letter:

4E-BP1 is a negative regulator of cap-dependent translation, and the phosphorylation of 4E-BP1 by mTOR or CDK1 inactivates 4E-BP1. We have now knocked down 4E-BP1 and monitored how this affects mitotic protein synthesis rates. Our new data shows that in the absence of 4E-BP1 CDK1 inhibition no longer reduces mitotic protein synthesis as radically. Thus, this rescue experiment validates that at least part of the CDK1 driven mitotic growth is mediated by 4E-BP1. Please see our new Figure 6G, Figure 6—figure supplement 4, and subsection “CDK1 drives phosphorylation of 4E-BP1, protein synthesis and mass accumulation in mitosis”.

We have clarified in our Discussion that other CDK1 dependent and independent growth regulations may also exist, as our data does not rule this out.

They should also discuss the potential role of mitotic swelling, which they showed is also CDK1 dependent, and could affect early mitotic growth rate – when cells enter mitosis, they increase volume in a CDK1 dependent manner. One would thus expect that inhibiting CDK1 activity, which would prevent this swelling, could have an effect on overall cell density and affect the growth rate in this early mitotic phase.

The reviewer raises another important concern. As stated in the major changes section at the beginning of our response letter:

We have now addressed how morphological changes (mitotic swelling, rounding and cytokinesis) affect our data. We have added completely new figures (Figure 5; Figure 5—figure supplement 1,2) to our manuscript, in which we study mitotic growth dynamics when we inhibit mitotic swelling. These experiments show that mitotic swelling is responsible for the increased MAR observed in early mitosis, but not for the rest of the MAR dynamics. Furthermore, mitotic protein synthesis dynamics did not display any change when swelling was inhibited. Please see subsection “Cells in metaphase and anaphase display mitotic stage specific inhibition of mass accumulation” for full details.

Notably, as we show in our previous work (Son et al., 2015), partial CDK1 inhibition with 1μM RO-3306 (same concentration used in this work) does not significantly change the magnitude of mitotic swelling in L1210 cells. Despite this, the partial CDK1 inhibition removes the increased growth observed in control cells after mitotic entry, suggesting that CDK1 inhibition results in reduction of growth that is independent of mitotic cell swelling.

2) One consequence of the previous concern is that the last piece of data provided is not really easy to interpret. It is in fact unlikely that the effect of CDK1 inhibition on the asymmetry of division is due to its effect on mitotic growth. This effect of CDK1 inhibition on mitotic spindle positioning, or in fact of almost any drug inducing a mitotic delay, as already been reported and rather depends on the importance of CDK activity for other processes that are needed to orient the spindle properly (see for example Beamish JBC 2009). I would remove this part, which does not bring much to the article and is unlikely to be correct – it is probably not because of a lack of early mitotic growth that the division ends up being asymmetrical upon CDK1 inhibition. In my opinion, the conclusion drawn “Together, these results show that CDK1 and the maintenance of active translation in M-phase are required for the fidelity of cytokinesis” is not supported by the data. But it is really not important for the rest of the article.3) Similarly, I would remove the very short chapter on the localization of 4E-BP1. It is not clear how it helps the main message of the article.

Thank you for pointing out these concerns. As suggested by the editors, we have now removed our last main figure and all data relating to the asymmetric cell division and the localization of p-4E-BP1.

4) To help the reader estimate the global effect of inhibiting early mitotic growth, the authors should provide, just like in Figure 1, the effective growth in mitosis (in% and not just the MAR), cytokinesis and early G1 comparing control and cells in which early mitotic growth has been inhibited.

Thank you for the suggestion. However, our experimental approach does not reveal where G1 ends. Thus, we cannot reveal the total growth during G1, only the growth rate at the beginning (first 30min) of G1. In order to do what the reviewer is suggesting, we would have to fix the duration analyzed for G1. As also the duration of early mitosis is relatively constant (Araujo et al., 2016), the early mitosis and G1 phases would display results comparable to our current MAR analysis (please see Author response image 2). We believe this would not provide additional information to the reader and therefore we do not present it in our manuscript.

**Author response image 2. respfig2:** Comparison of cell growth in early mitosis, cytokinesis and early G1 after indicated drug treatments. Normalized mass accumulation rate is displayed on top, while total mass accumulated is displayed on the bottom. Both formats of data presentation lead to the same conclusion.

[Editors' note: further revisions were requested prior to acceptance, as described below.]

Essential revisions:The 4E-BP1 knockdown experiment does not really prove that translation in mitosis is needed for growth in G1. One could still envision a scenario in which CDK1 inhibition causes some sort of stress in G1 phase, either through an increased mitotic duration or for example through induction of chromosome missegregation (or DNA damage associated with chromosome missegregation). Such stress could trigger a translational downregulation in G1, which is (partially) restored by depletion of 4E-BP1. In this scenario it is not the role of CDK1 in mitotic translation / growth that is responsible for slower G1, but another function of CDK1. Also, the depletion of 4E-BP1 does not increase G1 translation through its activity in M phase, but through its activity in G1 phase in this scenario.The authors should discuss this alternative possibility if they cannot fully exclude it.

We would like to thank the reviewers and editors for pointing out that alternative explanations exist that might explain how CDK1 inhibition affects daughter cell growth. We agree with the criticism and we have revised the results and Discussion section accordingly. First, in the Results section, we now state that “Thus, 4E-BP1 mediates mitotic protein synthesis of CDK1 (Figure 6), and this may also affect daughter cell protein synthesis. However, we cannot fully exclude the possibility that daughter cell growth is affected by some other effects of partial CDK1 inhibition, such as chromosome missegregation, which could consequently reduce daughter cell growth independently of mitotic growth.” In addition to this, we have also amended our Discussion section so that “CDK1 activity”, not necessarily “CDK1-driven mitotic growth”, affected daughter cell growth. We also state that “CDK1 inhibition may reduce daughter cell growth independently of mitotic growth”.